# Human-Plant Coevolution: A modelling framework for theory-building on the origins of agriculture

Andreas Angourakis[1,2]*, Jonas Alcaina-Mateos[3], Marco Madella[3,4,5], Debora Zurro[6]

**1** McDonald Institute for Archaeological Research, University of Cambridge, Cambridge, United Kingdom, **2** Institut für Archäologische Wissenschaften, Ruhr Universität Bochum, Bochum, Germany, **3** CaSEs, Department of Humanities, Universitat Pompeu Fabra, Barcelona, Spain, **4** Institució Catalana de Recerca i Estudis Avançats (ICREA), Barcelona, Spain, **5** School of Geography, Archaeology and Environmental Studies, The University of the Witwatersrand, Johannesburg, South Africa, **6** HUMANE - Human Ecology and Archaeology Research Group, Departamento de Arqueología y Antropología, Institución Milá y Fontanals de Investigación en Humanidades - Consejo Superior de Investigaciones Científicas (CSIC), Barcelona, Spain

☯ These authors contributed equally to this work.
\* aa2112cantab.ac.uk, andros.spica@gmail.com

**Data Availability Statement:** All data files associated to this article can be found in: Andreas Angourakis. (2021). Andros-Spica/hpcModel: Human-Plant Coevolution model: source files, simulation interface, sensitivity analysis report and

## Abstract

The domestication of plants and the origin of agricultural societies has been the focus of much theoretical discussion on why, how, when, and where these happened. The 'when' and 'where' have been substantially addressed by different branches of archaeology, thanks to advances in methodology and the broadening of the geographical and chronological scope of evidence. However, the 'why' and 'how' have lagged behind, holding on to relatively old models with limited explanatory power. Armed with the evidence now available, we can return to theory by revisiting the mechanisms allegedly involved, disentangling their connection to the diversity of trajectories, and identifying the weight and role of the parameters involved. We present the Human-Plant Coevolution (HPC) model, which represents the dynamics of coevolution between a human and a plant population. The model consists of an ecological positive feedback system (mutualism), which can be reinforced by positive evolutionary feedback (coevolution). The model formulation is the result of wiring together relatively simple simulation models of population ecology and evolution, through a computational implementation in R. The HPC model captures a variety of potential scenarios, though which conditions are linked to the degree and timing of population change and the intensity of selective pressures. Our results confirm that the possible trajectories leading to neolithisation are diverse and involve multiple factors. However, simulations also show how some of those factors are entangled, what are their effects on human and plant populations under different conditions, and what might be the main causes fostering agriculture and domestication.

documentation (v1.3). Zenodo. https://doi.org/10.5281/zenodo.6759456; 10.5281/zenodo.3881915 (all versions)

**Funding:** This research is part of the activities of the Culture and Socioecological Dynamics Research Group (CaSEs), a Quality Group of the Generalitat de Catalunya (2017 SGR212) (JAM, MM, DZ), and was supported by the CULM project (HAR2016-77672-P), funded by the former Spanish Ministry of Economy and Competitiveness (MINECO) (DZ); the projects TwoRains, funded by the European Research Council (ERC) under the European Union's Horizon 2020 research and innovation programme (grant agreement no 648609), and "Resources in Transformation" (ReForm) - Leibniz-WissenschaftsCampus, funded by Leibniz Gemeinschaft (AA). Previous development was also made possible through the support of the SimulPast Project— Consolider Ingenio 2010 (CSD2010-00034), funded by the former Spanish Ministry for Science and Innovation (MICIN) (AA, JAM, MM, DZ), and the CAMOTECCER project (HAR2012-32653) and FPI contract (BES-2013-062691), funded by MINECO (AA).

**Competing interests:** The authors have declared that no competing interests exist.

# Introduction

The domestication of plants and the origin of agriculture is a major change in human history, and it has been the focus of much theoretical discussion on why, how, when and where this change happened. Evidence from archaeobotany and plant genomics gathered during the last two decades expanded our knowledge on where this process happened and identified several centres of agricultural origin around the world [1–3]. Methodological advances in identification criteria [4] and the widespread recovery of plant remains from archaeological sites [5] better clarified the timing of this process in many areas. However, a better understanding of the why and how agriculture began seems to be still elusive [6–8].

Climate change [9–11], cognitive/symbolic change [12–14], or social competition and demography [15, 16] have long been discussed as drivers for socio-ecological transformations called the Neolithic Revolution [17]. A major problem with these approaches is to bundle under the same explanation behavioural trajectories that do not necessarily share the same premises. Domestication and agriculture emerged from diverse historical contexts and the empirical record available is manifold, inherently biased and fragmentary due to preservation issues, and it can often also be contradictory in evidencing causality [18]. Furthermore, several models rely on ethnographic observations of contemporary traditional practices among indigenous peoples around the world [19–23]. While these practices make a useful basis for creating models of the past, they may greatly differ in context from those of the first communities engaging in agriculture within any given region, and therefore such "parallelisms" need to be used with care [24].

A current and lively discourse on how domestication (and eventually agriculture) came into being is that of protracted [25–28] versus expedite [14, 29] domestication. Broad contextual analyses of the archaeobotanical record within macroevolutionary theory [18] and single-crop approaches [30] started to bring new light on the process of domestication based on a fast-growing body of archaeological evidence. The analysis of this massive and relatively recent volume of data makes clear that it is now necessary to return to theory by revisiting the mechanisms allegedly involved in domestication, disentangling their connection to a diversity of trajectories [31, 32], being those protracted or sudden, and identifying the weight of the social and ecological parameters. Approaches developed within human behavioural ecology [7, 33–38], such as niche construction or cultural niche construction theories, have gained momentum in this effort. These approaches emphasise "the capacity of organisms to modify natural selection in their environment and thereby act as co-directors of their own, and other species' evolution" [39]. However, such perspectives have been heavily criticised, especially as they are considered by some researchers indifferent to the role of human agency and intentionality [14, 29, 40, 41]. The relevant, yet stale, century-long debate on human intentionality in plant domestication is a clear sign that the field still lacks a unifying theoretical framework.

## Simulation approaches to human-plant coevolution

The study of the prehistoric human past is necessarily approached through the archaeological record, which does not always allow addressing historical processes and organizational dynamics. Information gaps as well as uncertainty in the record are behind the push for archaeology to participate in and take advantage of innovative methodological approaches, such as modelling and simulation. In a subject like domestication and the origins of agriculture, where the archaeological record is incomplete in both space and time, and real-world experiments are unrealistic, the use of modelling and simulation has become a useful alternative for testing hypotheses and building theory [42]. However, the most important contributions within this framework have focus on the representation of plant domestication in terms

of genetic change [25, 43] and the geographical spread of the Neolithic transition [44–47], mainly for testing hypotheses related to regional or species-wise case studies. Exceptionally, there have been key contributions from niche construction and optimal foraging theory as well as complex adaptative systems, but such contributions have been mostly centred on the human side of the process [31, 38, 48–50]. Few simulation models have considered coevolution as the core mechanism producing changes in both plants and humans [51, 52], while the first proposals in this line date back to almost forty years ago [53].

The current work explores hypotheses on plant domestication and the origin of agriculture by using a coevolutionary framework capable of accounting for both plant and human factors. Our model combines readily-available formal models for mutualism and evolution used in population ecology, sociology and economics. Despite sharing the term "coevolution", our approach is neither based on nor necessarily aligned with the gene-culture coevolution or dual inheritance theory. The latter concerns a coupled process of genetic and cultural change in the same population and species, typically humans and other primates, in which other populations and species, and their changes, are considered as factors rather than the subjects of coevolution [54]. Likewise, the model we propose can be distinguished from human behaviour ecology models in this field since these have been defined in terms of human behaviour only (e.g., focusing on decision-making criteria) while factoring other species primarily as resources [38, 55].

We state our model assumptions explicitly and have worked intensively on documenting all implementation details to assure its reproducibility and facilitate re-use and future expansions. Our contribution is theoretical and explorative, thus it is not driven by the use of any specific dataset or case study. Furthermore, it does not carry the pretence —at least in its current form— of direct applicability to the many formats of empirical data.

## The Human-Plant Coevolution (HPC) model

Human-plant interaction is a specific case of animal-plant interaction, which spans predator-prey, mutualistic and symbiotic relationships. All ecological relationships consistent in time are driven by coevolution, where each party exerts selective pressures on the other, eventually redefining their genetic (and cultural) construct [53, 56–58]. Under mutualistic coevolution, the interaction between two populations increases the total potential return or carrying capacity of the environment for each species. At the same time, it also modifies the selective pressures acting over the populations involved. In this light, plant domestication is similar to other mutualistic relationships, where coevolution made possible the emergence of certain traits, manifested at physiological, morphological and behavioural levels; e.g., insects and fungi [59] and ants and acacias [60].

The Human-Plant Coevolution (HPC) model is an ecological positive feedback system (mutualism), which can be reinforced by an evolutionary positive feedback (coevolution). The model is the result of wiring together relatively simple models of population ecology (Verhulst-Pearl model) and evolution (replicator dynamics), through a computational implementation using R programming language [61].

The HPC model embodies the dynamics of two interacting populations: one of humans and another of a given plant species. Here, we assume that population units are individual organisms. Because this model greatly simplifies the mechanisms involved in population dynamics, units could also be set to be groups of individuals or even population proxies (e.g. human working hours, plant-covered soil surface). However, the scale of population units is relevant when calibrating the parameters and interpreting results, and thus must be made explicit.

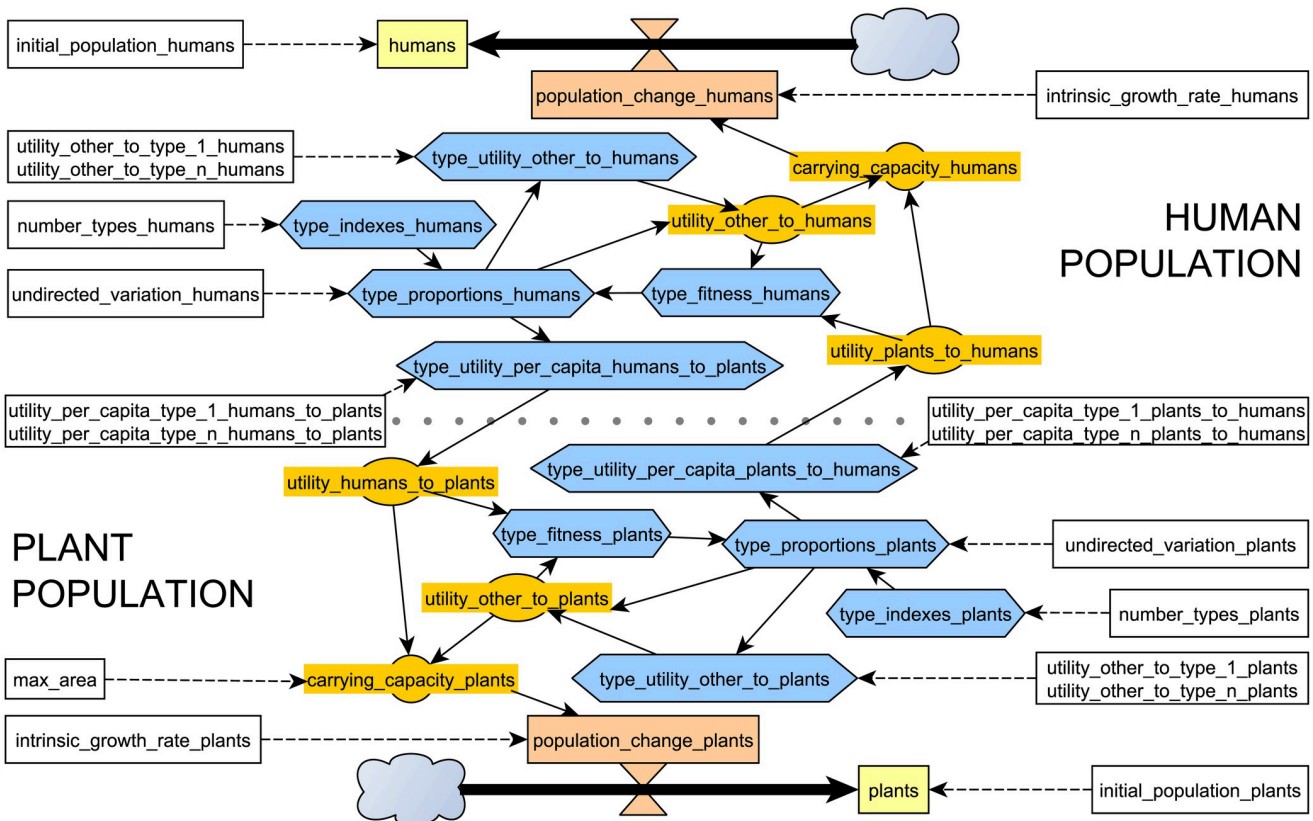

**Fig 1. Simplified forrester diagram representing the relationships between parameters and main variables of the Human-Plant Coevolution (using R notation; see Tables 1 and 2).** Populations are shown in yellow, their change in red, type-wise vector or array variables in blue, aggregate population variables in orange, and parameters in white.

Each population unit may exploit the available resources in different ways, and may have a different utility for sustaining the other population. To represent this, we assume that each population can be divided into types ranging from the least (1) to the most (n) mutualistic, each corresponding to a value of baseline carrying capacity and utility per capita, which in turn range from population-specific minima and maxima. Each type can relate either to truly discrete units (e.g., presence/absence of trait), arbitrary degrees in a continuum (e.g., size of anatomy trait, frequency of behaviour), or a combination of both. In the case of human populations, types would consist majorly of different combinations of behaviours impacting the plant population, such as protection from predators, removal of competitors, enhancement of soil conditions, or transporting and storing propagules.

This simplification of population diversity gives the possibility to implement a relatively simple and straightforward mechanism of evolution, the replicator dynamics [62]. Under our specific version of this mechanism, the distribution of a population within types changes depending on three factors: undirected variation, inertia, and selection.

The HPC model was conceptualised as a highly symmetric structure (Fig 1). This model reduces the complexity of the human and plant populations to a point where these can be defined using the same terms (parameters and variables). The symmetry is only broken by the inclusion of a constraint specific to plants, the maximum number of plant units fitting

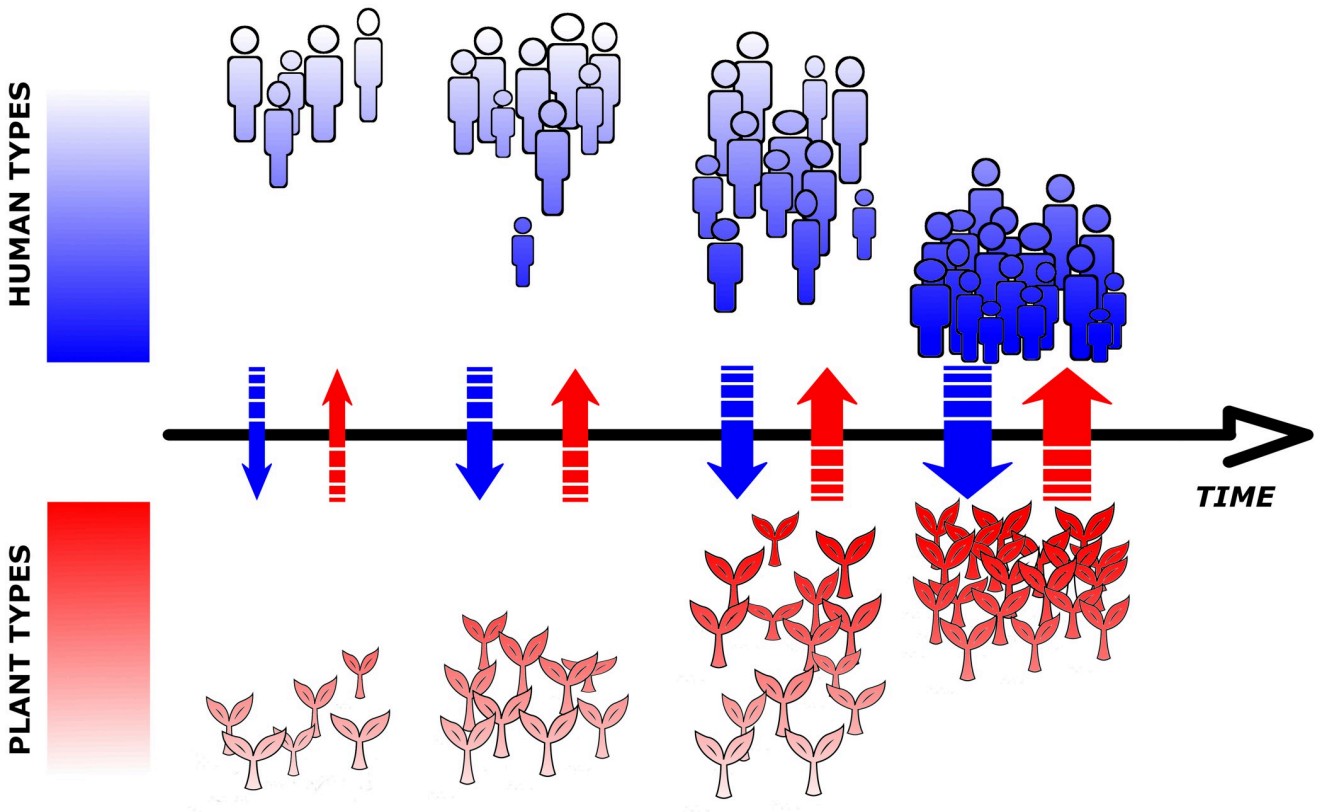

**Fig 2. A successful case of coupled mutualism and coevolution, as defined in the Human-Plant Coevolution model.** As the interaction between populations (coloured arrows) becomes stronger, carrying capacities increase and populations grow (number of organisms) and stronger mutualism types (stronger colour shades) become more frequent.

the area available (MaxArea or `max_area`), reflecting one of the main ecological differences between plants and animals: the latter are able to move and exploit multiple habitats within a lifetime.

The HPC model enables to reproduce a double positive feedback loop, where two populations increase their carrying capacity (mutualism) and empower this relationship by influencing each other's trait selection (coevolution). The consequence is that, given certain conditions, both human and plant populations shift to stronger mutualism types and increase their numbers, potentially moving far away from pre-coevolutionary levels (Fig 2).

All parameters and variables of the model are listed and defined in Tables 1 and 2, respectively. States of the system are evaluated and compared by a set of output variables, i.e. those not used to recalculate the state of the system (Table 3). Among the output variables, the coevolution coefficients are the most revealing. Each indicates if and how much the population type distribution has been modified by the coevolutionary process. Their values range between -1 (the entire population is of type 1) and 1 (the entire population is of type n).

## Ecological relationships and population dynamics

The model can be expressed by a relatively simple system of two discrete-time difference equations Eq (1), based on the Verhulst-Pearl Logistic equation [63, 64]. The change of both

**Table 1. Parameters.**

| R notation | Math. notation | Description |
|---|---|---|
| `initial_population_humans,`<br>`initial_population_plants` | $ini_H$, $ini_P$ | Initial populations of humans and plants |
| `number_types_humans, number_types_plants` | $n_H$, $n_P$ | Number of types of humans and plants |
| `undirected_variation_humans,`<br>`undirected_variation_plants` | $v_H$, $v_P$ | Level of undirected variation in humans and plants |
| `intrinsic_growth_rate_humans,`<br>`intrinsic_growth_rate_plants` | $r_H$, $r_P$ | Intrinsic growth rates for human and plant populations |
| `utility_per_capita_type_n_plants_to_humans` | $\bar{U}_{P_nH}$ | Utility per capita of type n plants to humans |
| `utility_per_capita_type_1_plants_to_humans` | $\bar{U}_{P_1H}$ | Utility per capita of type 1 plants to humans |
| `utility_per_capita_type_n_humans_to_plants` | $\bar{U}_{H_nP}$ | utility per capita of type n humans to plants |
| `utility_per_capita_type_1_humans_to_plants` | $\bar{U}_{H_1P}$ | Utility per capita of type 1 humans to plants |
| `utility_other_to_type_n_plants` | $U_{bPn}$ | Utility of other resources to type n plants |
| `utility_other_to_type_1_plants` | $U_{bP1}$ | Utility of other resources to type 1 plants |
| `utility_other_to_type_n_humans` | $U_{bHn}$ | Utility of other resources to type n humans |
| `utility_other_to_type_1_humans` | $U_{bH1}$ | Utility of other resources to type 1 humans |
| `max_area` | MaxArea | Maximum number of plant population units fitting the contiguous area available |
| `max_iterations` | $time_{max}$ | Maximum number of iterations allowed before halting a simulation run |
| `reltol_exponential` | $\epsilon$ | Base 10 negative exponential controlling how small population change must be to halt a simulation run |
| `coevolution_threshold` | $coevo_\theta$ | Value between -1 and 1 to which to compare coevolution coefficients and decide if qualitative shift in type proportions has happened, so timing can be registered (see also Table 3) |

populations ($\Delta_H[t]$ or `population_change_humans`, $\Delta_P[t]$ or `population_chan-ge_plants`; see Table 2) depends on an intrinsic growth rate ($r_H$ or `intrinsic_grow-th_rate_humans`, $r_P$ or `intrinsic_growth_rate_plants`), the population at a given time ($H[t]$ or `humans`, $P[t]$ or `plants`) and the respective carrying capacity of the

**Table 2. Variables.**

| R notation | Math. notation | Description |
|---|---|---|
| `humans, plants` | $H[t]$, $P[t]$ | Human and plant populations |
| `carrying_capacity_humans, carrying_capacity_plants` | $K_H[t]$, $K_P[t]$ | Carrying capacity to human and plant populations |
| `utility_humans_to_plants, utility_plants_to_humans` | $U_{HP}[t]$, $U_{PH}[t]$ | Utility of one population to the other |
| `utility_other_to_humans, utility_other_to_plants` | $U_{bH}[t]$, $U_{bP}[t]$ | Utility of other resources to a population (baseline carrying capacity) |
| `type_indexes_humans, type_indexes_plants` | $types_H$, $types_P$ | Population types, arbitrarily ordered from 1 to n (vector) |
| `type_proportions_humans, type_proportions_plants` | $pop_H[t]$, $pop_P[t]$ | Proportion of a population belonging to type i (vector) |
| `type_utility_per_capita_humans_to_plants,`<br>`type_utility_per_capita_plants_to_humans` | $U_{HP}$, $U_{PH}$ | Utility per capita of type i individuals of one population to (average) individuals in the other (vector) |
| `type_utility_other_to_humans, type_utility_other_to_plants` | $U_{bHi}$, $U_{bPi}$ | Utility of other resources to type i individuals of a population (vector) |
| `type_fitness_humans, type_fitness_plants` | $fitness_H[t]$,<br>$fitness_P[t]$ | Fitness score of individuals of each type in a population (vector) |
| `population_change_humans, population_change_plants` | $\Delta_H[t]$, $\Delta_P[t]$ | Population change at time t (vector) |

**Table 3. Variables (output only).**

| R notation | Math. notation | Description |
|---|---|---|
| `coevolution_coefficient_humans,` `coevolution_coefficient_plants` | $coevo_H[t]$, $coevo_P[t]$ | Coevolution coefficient or the distribution of the proportions of a population per type weighted by type index |
| `dependency_coefficient_humans,` `dependency_coefficient_plants` | $depend_H[t]$, $depend_P[t]$ | Dependency coefficient or the slope of the linear model of the fitness score per type (e.g., from $fitness_{H1}[t]$ to $fitness_{Hn}[t]$) using type index (1 to $n_H$) |
| `timing_humans, timing_plants` | $timing_H$, $timing_P$ | Iterations past until coevolution successfully changes the proportions of population per type |
| `time_end` | $t_{end}$ | Iterations past until the end-state |

environment for each population ($K_H[t]$ or `carrying_capacity_humans`, $K_P[t]$ or `carrying_capacity_plants`), which may also vary over time.

$$H[t+1] = H[t] + r_H H[t] - r_H \frac{H[t]^2}{K_H[t]} \tag{1a}$$

$$P[t+1] = P[t] + r_P P[t] - r_P \frac{P[t]^2}{K_P[t]} \tag{1b}$$

Human and plant populations engage in a mutualistic relationship, where one species is to some extent sustained by the other Eq (2). The mutualistic relationship is defined in the model as an increment of the carrying capacity of one population caused by the other. The increment in each population, expressed as the utility at a given time of humans to plants ($U_{HP}[t]$ or `utility_humans_to_plants`) and plants to humans ($U_{PH}[t]$ or `utility_plants_to_humans`), is the product of the utility per capita of individuals in one population to individuals in the other ($\bar{U}_{HP}[t]$, $\bar{U}_{PH}[t]$) and the number of individuals in the utility-giving population ($H[t]$ or `humans`, $P[t]$ or `plants`) Eq (3).

Both populations are also sustained by an independent term, representing the baseline carrying capacity of the environment or the utility gain from other resources, which is time-dependent ($U_{bH}[t]$ or `utility_other_to_humans`, $U_{bP}[t]$ or `utility_other_to_plants`). While assuming that the growth of the human population has no predefined ceiling, the expansion of the plant population is considered limited by the area over which plants can grow contiguously (MaxArea or `max_area`), and represented as a compendium of both space and the maximum energy available in a discrete location Eq (2a).

$$K_H[t] = U_{PH}[t] + U_{bH}[t] \tag{2a}$$

$$K_P[t] = \min(U_{HP}[t] + U_{bP}[t], MaxArea) \tag{2b}$$

$$U_{HP}[t] = H[t] \cdot \bar{U}_{HP} \tag{3a}$$

$$U_{PH}[t] = P[t] \cdot \bar{U}_{PH} \tag{3b}$$

Considering that mutualistic relationships involve a positive feedback loop, the population growth at time t improves the conditions for both humans and plants at time t + 1, sustaining their growth even further. See model assumptions in Table 4.

**Population diversity.** The HPC model contemplates a vector pop of length n for each population, containing the population fractions of each type ($pop_H[t]$ or `type_proportions_humans`, $pop_P[t]$ or `type_proportions_plants`). The lengths of these vectors

**Table 4. Assumptions on ecological relationships and population dynamics.**

| Domains | Assumptions |
|---|---|
| On interacting populations | A population of humans interacts with a population of plants. |
| On population growth | Population growth is a self-catalysing process, where the population density in the present will contribute to its own increase in the future, depending on an intrinsic growth rate (r).<br>Population growth is a self-limiting process, where the population density in the present will constraint its own increase in the future, depending on respective carrying capacity of the environment (K).<br>The logistic growth model is acceptable as an approximation to the dynamics of populations, both human and plant, under constant conditions.<br>The carrying capacity of the environment for a population depends on constant factors and on a time-varying factor (K[t]). |
| On positive ecological relationships | Positive ecological relationships exist, where an individual of one population increases by an amount the carrying capacity of the environment for another population.<br>Coupled positive ecological relationships (i.e., mutualism) exist, where two populations increase the carrying capacities for each other.<br>There is variation in positive ecological relationships, so individuals of one population vary in terms of how much they increase the carrying capacity for the other population. |
| On human-plant mutualism | A given plant species yield a positive utility for humans, e.g., as a source of food and raw materials.<br>Humans return a positive utility for this plant species, e.g., by improving soil conditions.<br>The utility given by one population adds value to the carrying capacity for the other, and vice versa.<br>The carrying capacity for humans rely also on other resources, which are independent of the plant species (i.e., the baseline carrying capacity for humans).<br>The carrying capacity for plants also relies on other conditions, which are independent of humans (i.e., the baseline carrying capacity for plants).<br>The carrying capacity for plants is eventually constrained by the space available for it to grow contiguously as a population (i.e., maximum area). |

or the numbers of types are population-specific and given as two parameters ($n_H$ or `number_types_humans`, $n_P$ or `number_types_plants`). These vectors include all possible variations within a population so that they each amount to unity (i.e. $\sum_{i=1}^{n} pop_{H_i} = 1$ and $\sum_{i=1}^{n} pop_{P_i} = 1$).

To account for multiple types, we replace Eq (3) with Eq (4), where the utility of one population to the other at any given time ($U_{HP}[t]$ or `utility_humans_to_plants`, $U_{PH}[t]$ or `utility_plants_to_humans`) is calculated by summing up the utility per capita of each type ($\bar{U}_{H_iP}[t]$ or `type_utility_per_capita_humans_to_plants`, $\bar{U}_{P_iH}[t]$ or `type_utility_per_capita_plants_to_humans`) proportionally to the share of population of the respective type ($pop_H[t]$ or `type_proportions_humans`, $pop_P[t]$ or `type_proportions_plants`), and multiplying the result by the population size ($H[t]$ or `humans`, $P[t]$ or `plants`). The baseline carrying capacities ($U_{bH}[t]$ or `utility_other_-to_humans`, $U_{bP}[t]$ or `utility_other_to_plants`) are calculated similarly, though using the utility that each type is able to gain from other resources ($U_{bHi}$ or `type_utility_other_to_humans`, $U_{bPi}$ or `type_utility_other_to_plants`) Eq (5).

$$U_{HP}[t] = H[t] \sum_{i=1}^{n_H} pop_{H_i}[t] \cdot \bar{U}_{H_iP} \tag{4a}$$

$$U_{PH}[t] = P[t] \sum_{i=1}^{n_P} pop_{P_i}[t] \cdot \bar{U}_{P_iH} \tag{4b}$$

$$U_{bH}[t] = \sum_{i=1}^{n_H} pop_{H_i}[t] \cdot U_{bH_i} \tag{5a}$$

$$U_{bP}[t] = \sum_{i=1}^{n_P} pop_{P_i}[t] \cdot U_{bP_i} \tag{5b}$$

Types relate to population-specific values of utility per capita ($\bar{U}_{H_iP}[t]$ or `type_utili-ty_per_capita_humans_to_plants`, $\bar{U}_{P_iH}[t]$ or `type_utility_per_capita_-plants_to_humans`) and baseline carrying capacity ($U_{bH}[t]$ or `utility_other_to_humans`, $U_{bP}[t]$ or `utility_other_to_plants`). The values corresponding to each type are defined by linear interpolation between pairs of parameters representing the values corresponding to types 1 and n (e.g., if $n_P = 10$, $\bar{U}_{P_1H} = 1$ and $\bar{U}_{P_nH} = 10$, then $\bar{U}_{P_5H} = 5$). The shares of population within types follow a one-tail distribution rather than a normal distribution, which would be more adequate but less straightforward to use in a theoretical model. Under this circumstance, the distribution of population within types will always be biased towards the intermediate types.

**Coevolutionary dynamics.** Undirected variation, which causes part of the population to randomly change to other types, represents the effect of mutation in genetic transmission or of innovation, error, and other mechanisms in cultural transmission. The proportion of individuals of type i in a population at time t ($pop_{Hi}[t]$ or `type_proportions_humans[i]`, $pop_{Pi}[t]$ or `type_proportions_plants[i]`), *after undirected variation* ($pop_{Hi}[t]'$, $pop_{Pi}[t]'$), depends on the level of undirected variation in that population ($v_H$ or `undirec-ted_variation_humans`, $v_P$ or `undirected_variation_plants`) and on the degree and sign of the difference between the current number of individuals of type i ($pop_{Hi}[t]$, $pop_{Pi}[t]$) and the expected proportion per type, assuming a uniform distribution among types ($1/n_H$ and $1/n_P$) Eq (6).

$$pop_H[t]' = pop_H[t] + v_H\left(\frac{1}{n_H} - pop_H[t]\right) \tag{6a}$$

$$pop_P[t]' = pop_P[t] + v_P\left(\frac{1}{n_P} - pop_P[t]\right) \tag{6b}$$

By considering inertia as an evolutionary mechanism, we assume that the more frequent a type is, the more likely that it is transmitted. Selection is implemented by assigning a fitness score to each type (fitness$_{Hi}[t]$ or `fitness_humans`, fitness$_{Pi}[t]$ or `fitness_plants`), which in turn biases its transmission. Eq (7) summarizes the combined effect that inertia and selection have on the proportion of population belonging to type i ($pop_{Hi}[t]$ or `type_pro-portions_humans[i]`, $pop_{Pi}[t]$ or `type_proportions_plants[i]`). For a formal

similarity of the discrete replicator dynamic and Bayesian inference, see [65].

$$pop_{H_i}[t + 1] = \frac{fitness_{H_i}[t] \cdot pop_{H_i}[t]}{\sum_{j=1}^{n_H} fitness_{H_j}[t] \cdot pop_{H_j}[t]} \tag{7a}$$

$$pop_{P_i}[t + 1] = \frac{fitness_{P_i}[t] \cdot pop_{P_i}[t]}{\sum_{j=1}^{n_P} fitness_{P_j}[t] \cdot pop_{P_j}[t]} \tag{7b}$$

The replicator dynamics described so far defines how a trait evolves in a single population. However, coevolution can also be represented when the selective pressure on one population is modified by the changing traits of another population. In order to link two populations, the fitness scores of one population are derived from the weight of the contribution or utility of the other population ($U_{HP}[t]$ or `utility_humans_to_plants`, $U_{PH}[t]$ or `utility_plants_to_humans`) in relation to the base carrying capacity for the former ($U_{bH}[t]$ or `utility_other_to_humans`, $U_{bP}[t]$ or `utility_other_to_plants`) Eq (8).

$$fitness_{H_i}[t] = \frac{(n_H - i)U_{bH}[t] + iU_{PH}[t]}{U_{bH}[t] + U_{PH}[t]} \tag{8a}$$

$$fitness_{P_i}[t] = \frac{(n_P - i)U_{bP}[t] + iU_{HP}[t]}{U_{bP}[t] + U_{HP}[t]} \tag{8b}$$

As a consequence of the model design, types of both human and plant populations span from a non-mutualistic type ($i = 1$), which has the best fitness score when there is no positive interaction with the other population (e.g., type 1 plants when $U_{HP}[t] \approx 0$), to a mutualistic type ($i = n$), which is the optimum when nearly the whole of the carrying capacity is due to such relationship (e.g., $U_{HP}[t] \approx K_P[t]$). See model assumptions in Table 5.

## End-state condition

A simulation ends when both populations and their respective type distributions are stable; i.e. no further change occurs given current conditions. More specifically, we use the RelTol method to decide if the absolute difference between the populations between time t and t—1 is

**Table 5. Assumptions on population diversity and coevolution.**

| Domains | Assumptions |
|---|---|
| On the evolution of traits | A population can be divided into types according to one or more traits. The distribution of individuals among types can vary in time, due to factors affecting trait transmission. |
| On the factors affecting the evolution of traits | Change of the population distribution among types depends on the previous population distribution: the more frequent is a type, the more likely it will be imitated or transmitted to the next generation. Change of the population distribution among types depends on the relative fitness of types: the greater the fitness score associated to a type, the more likely it will be imitated or transmitted to the next generation. Change of the population distribution among types depends on undirected variation. |
| On the coevolution of traits related to human-plant mutualism | The utility given by an individual varies within types. The utility given by other resources to a population varies within its types. The fitness of human types is modified by the relative weight of plant utility in the carrying capacity for humans The fitness of plant types is modified to the relative weight of human utility in the carrying capacity for plants. |

very small, less than $10^{-\epsilon}$ where $\epsilon = 6$ in our default setting (see `reltol_exponential` in Table 1). End-states defined by unchanged variables are known as stationary points. Exceptionally, under certain parameter settings, the HPC model does not converge into a stationary point but enters an oscillatory state. To handle these rare cases and others producing extremely slow-paced dynamics, simulations are interrupted regardless of the conditions after $time_{max}$ iterations (`max_iterations`, in the implementation in R).

### Output variables

The most important output variables are the coevolution coefficients ($coevo_H[t]$ or `coevolution_coefficient_humans`, $coevo_P[t]$ or `coevolution_coefficient_plants`), which measure the trend in the distribution of a population among its types Eq (9).

$$coevo_H[t] = \frac{\sum_{i=1}^{n_H} pop_{H_i}[t] * (types_{H_i} - 1)}{n_H - 1} * 2 - 1 \tag{9a}$$

$$coevo_P[t] = \frac{\sum_{i=1}^{n_P} pop_{P_i}[t] * (types_{P_i} - 1)}{n_P - 1} * 2 - 1 \tag{9b}$$

The dependency coefficients ($depend_H[t]$ or `dependency_coefficient_humans`, $depend_P[t]$ or `dependency_coefficient_plants`) express the direction and intensity of the selective pressure caused by the other population. It is calculated as the slope coefficient of a linear model of the fitness scores ($fitness_{Hi}[t]$ or `fitness_humans`, $fitness_{Pi}[t]$ or `fitness_plants`) using the type indexes ($types_H$ or `type_indexes_humans`, $types_P$ or `type_indexes_plants`) as an independent variable.

Positive values of both these coefficients reflect the tendency of a population towards the most mutualistic types (effective coevolution), while negative values indicate an inclination towards the non-mutualistic type due to a low selective pressure exerted by the mutualistic relationship.

We recorded the time step at the end of each simulation ($time_{end}$ or `time_end`), obtaining a measure of the overall duration of the process. Whenever applicable, we registered the duration of change towards stronger mutualism types in both populations ($timing_H$ or `timing_humans`, $timing_P$ or `timing_plants`). We consider change to be effective when at least half of a population is in the higher quarter of the type spectrum, with the respective coevolution coefficient being greater than 0.5 in a scale from -1 to 1 ($coevo_\theta$ or `coevolution_threshold`).

### Experimental design

Although relatively simple, the HPC model has a total of 17 parameters. We did not engage in fixing any of these parameters to fit a particular case study as a strategy to reduce the complexity of results. In turn, as our aim is to theoretically explore human-plant coevolution, we scrutinised the 'multiverse' of scenarios that potentially represent the relationship between any given human population and any given plant species. The complexity of the model was managed by exploring the parameter space progressively, observing the multiplicity of cases in single runs, two and four-parameter explorations, and an extensive exploration including 15 parameters (all, except $ini_H$ and $ini_P$). The latter modality of exploration was performed by simulating 10,000 parameter settings sampled with the Latin Hypercube Sampling (LHS) technique [66] and Strauss optimization [67]. All simulation runs were executed for a maximum of 5,000 time steps, but most reached the end condition much earlier.

## Model implementation and additional materials

The source files associated with the HPC model are maintained in a dedicated online repository [68]: https://github.com/Andros-Spica/hpcModel. This repository contains several additional materials, including a web application to run simulations and the full report on the sensitivity analysis.

The Human-Plant Coevolution model can generate trajectories with or without the final occurrence of human-plant coevolution. Moreover, simulations revealed a broad spectrum of cases (Fig 3), including those where coevolution produces oscillatory or asymmetric change.

Throughout all conditions explored, the results show that a completely successful coevolutionary trajectory, where both populations effectively change, is relatively demanding and it can be deemed unlikely, considering the entirety of the parameter space explored. Furthermore, in light of these results, plant populations are systematically more sensitive to the selective pressure of mutualism than humans, arguing for the scarcity of cases of origins of agriculture in comparison to a relative abundance of effective domestication processes.

## End-states

The wide variety of end-states produced by the HPC model can be classified in three general groups:

- Coevolution does not occur. Simulation runs in which a stationary point is reached without successful coevolution, thus returning a stable state where humans and plants have a weak mutualistic relationship.

- Coevolution occurs. Both populations go through successful coevolution and become stable only once they have shifted towards stronger mutualism types.

- Coevolution occurs partially, encompassing two types of end-states:

  - Stationary suboptimal mutualism: One or both populations undergo a significant, but partial change, remaining relatively well distributed among types, or

  - Oscillatory coevolution: Both populations become trapped in an endless cycle alternating engagement (strong mutualism) and release (weak mutualism).

**Coevolution does not occur.** Under some conditions, equilibrium is reached without coevolution taking place and consequently both human and plant populations are kept at relatively low densities (Fig 4). Without coevolution, the plant population exists mainly in the non-anthropic niche ($U_{bP} \gg U_{HP}$, or `utility_other_to_plants` is much greater than `utility_humans_to_plants`) and in wild forms ($pop_{P_1} \gg pop_{P_n}$, or `type_proportions_plants[1]` is much greater than `type_proportions_plants[number_types_plants]`), while the bulk of human subsistence comes from other resources and only marginally from gathering these plants ($U_{bH} \gg U_{PH}$, or `utility_other_to_humans` is much greater than `utility_plants_to_humans`), which most humans do opportunistically and with little impact ($fitness_{P_1} \gg fitness_{P_n}$, or `fitness_plants[1]` is much greater than `fitness_plants[number_types_plants]`). End-states of this type can still diverge significantly due to different parameter settings.

**Coevolution occurs.** As intended, the HPC model is able to generate trajectories where equilibrium is reached with coevolution, and mutualism between humans and plants is reinforced (Fig 5; Animation 2). The plant population relies more on the human contribution ($U_{bP} \ll U_{HP}$, or `utility_other_to_plants` much less than

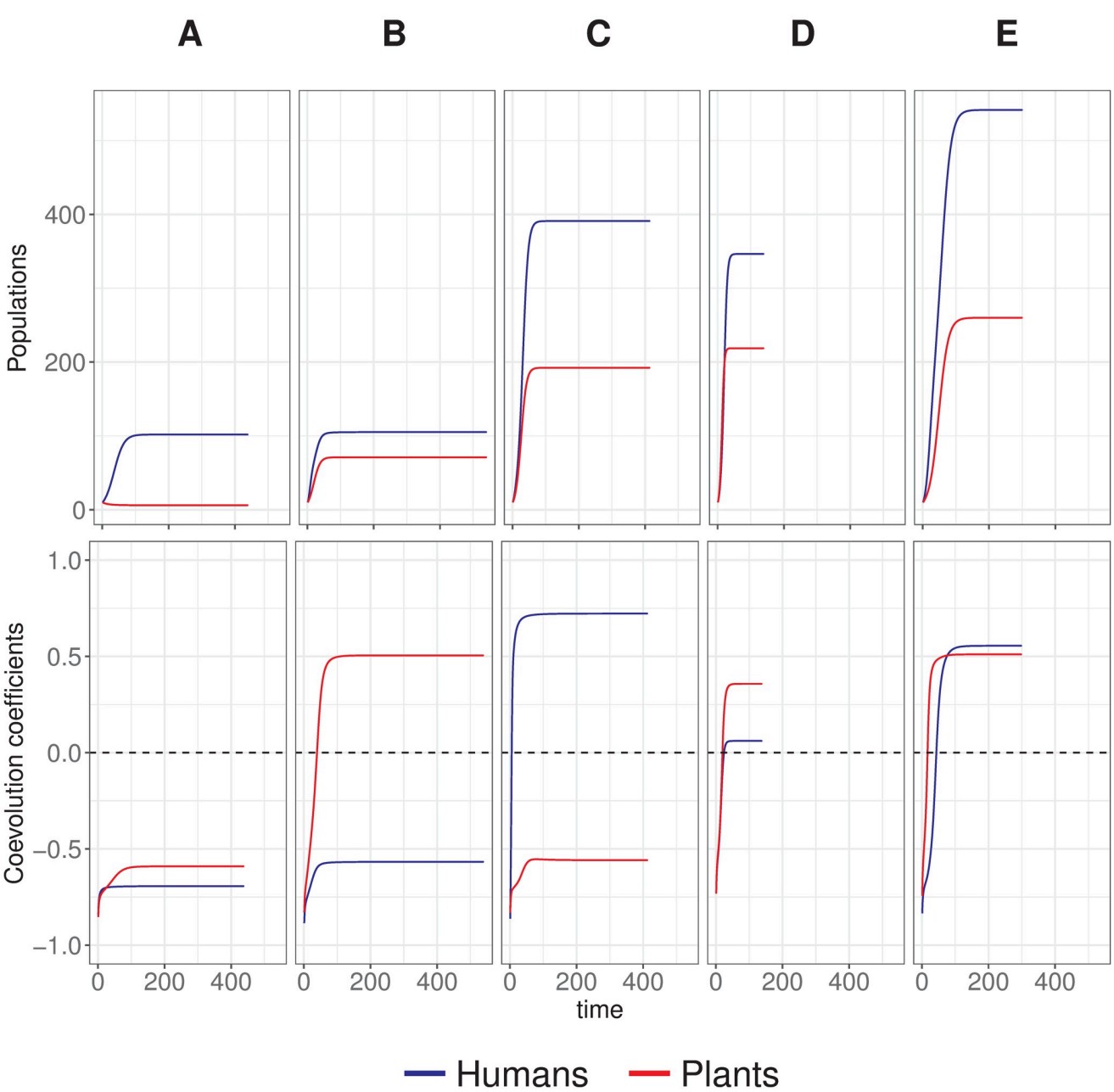

**Fig 3. Examples of trajectories and end-states produced by the Human-Plant Coevolution model.** A: no coevolution; B: only plant population changes (domestication without cultivation); C: only human population changes (cultivation without domestication); D: some change happens in both populations (diverse populations); E: strong change in both populations (domestication and cultivation). More details on the timing of changes are given in the following sections.

`utility_humans_to_plants`) and humans depend significantly on harvesting these plants ($U_{bH} \ll U_{PH}$, or `utility_other_to_humans` much less than `utility_plants_to_humans`).

As a general rule, the coevolved human and plant populations reach higher levels compared to their counterparts in non-coevolutionary end-states under similar conditions. The total contribution from one population to the other will increase when coevolution happens,

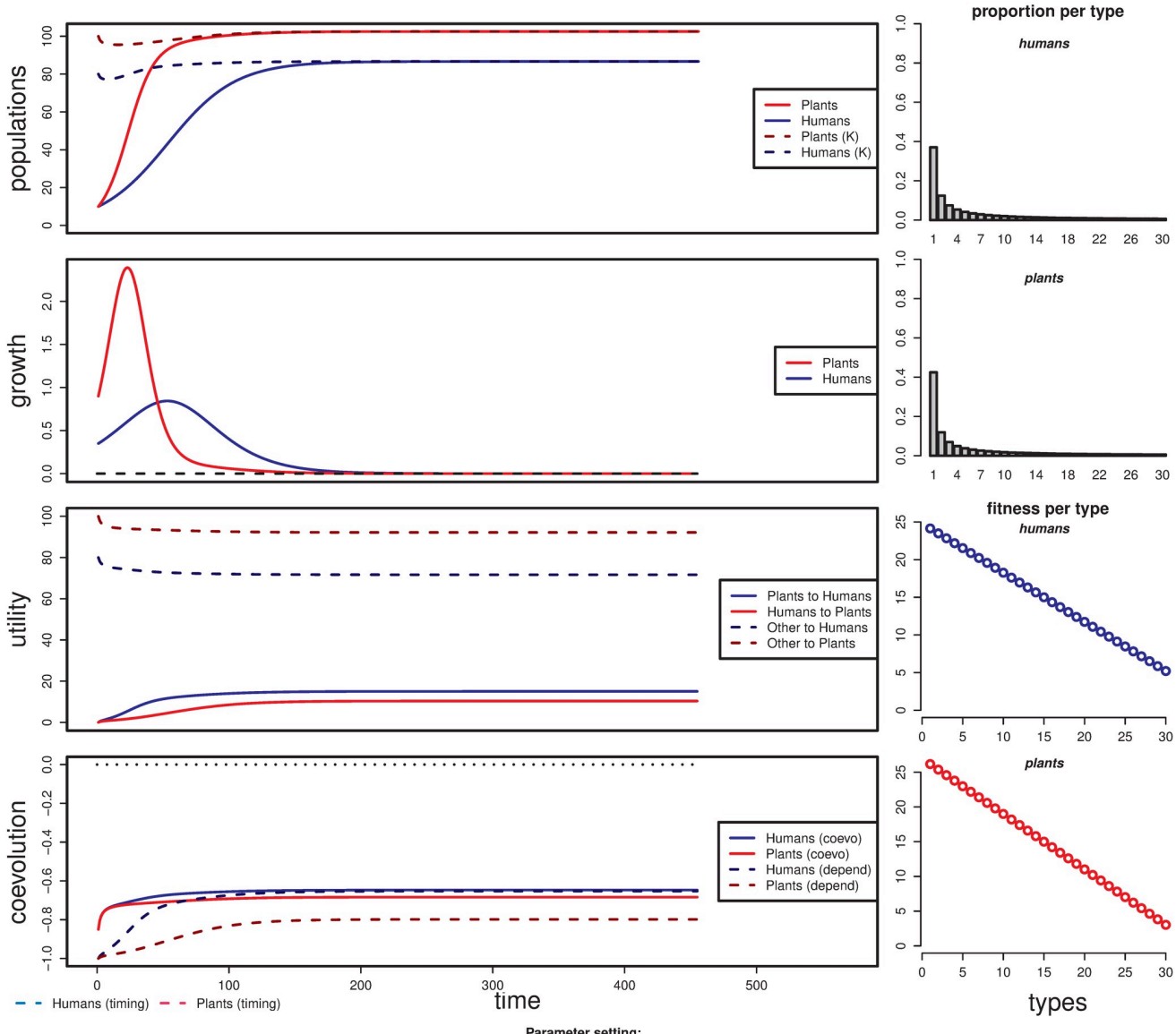

**Fig 4. Example of a simulation run producing a trajectory without coevolution.** The dynamics is reduced to the changes required for the initial populations to adjust their levels and type distribution to the least mutualistic stable state under the given conditions.

because of the positive feedback loop between population numbers: i.e. the more humans, more plants, and vice-versa.

In most cases where coevolution happens, the difference between the pseudo-stable and stable population levels before and after coevolution is fairly clear. These two levels are visible as the first and second plateaus in the double-sigmoid curve (see population plot in Fig 5, top left). The steep slope that mediates between these two levels follows the change in the

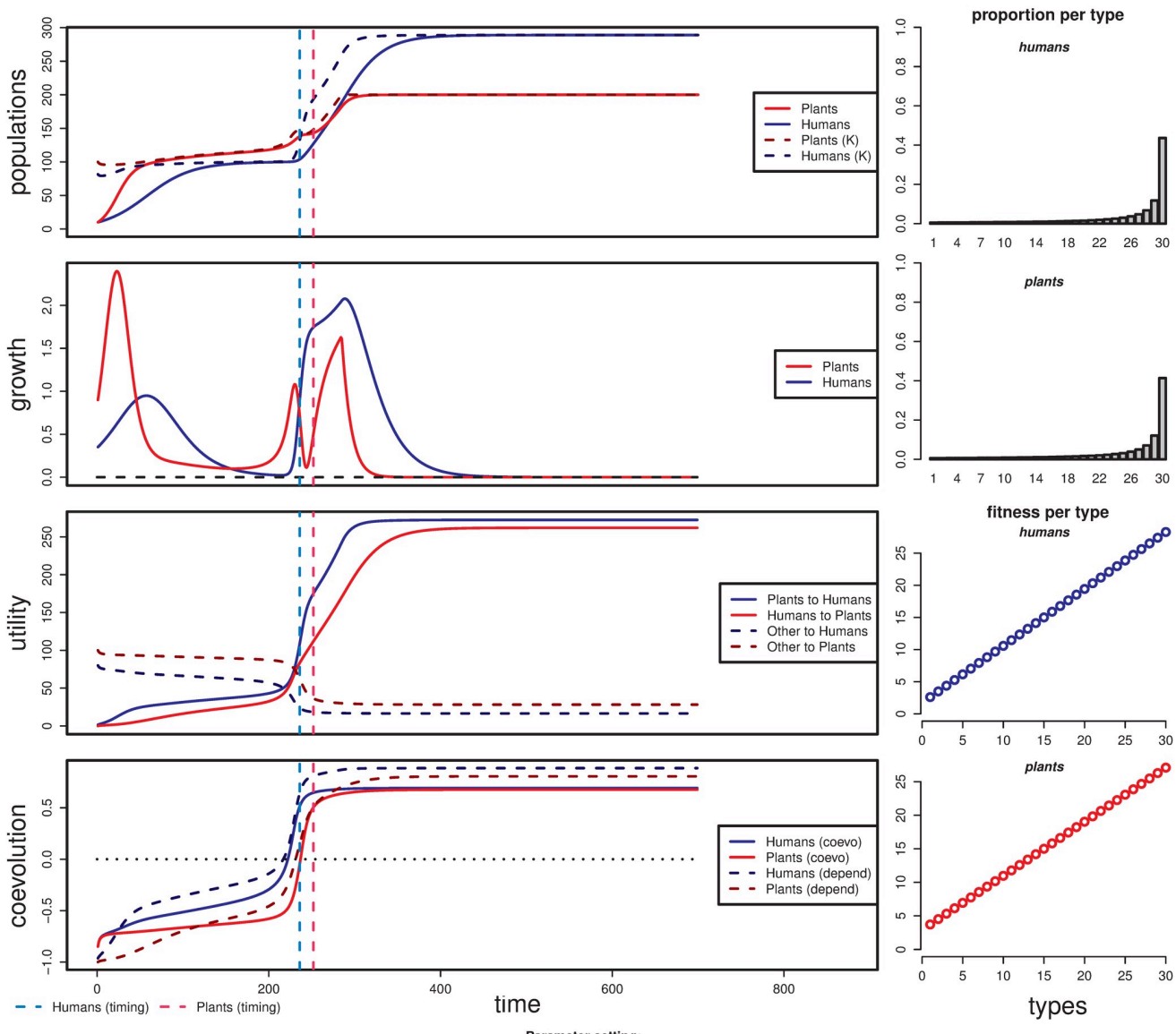

**Fig 5. Example of a simulation run with a case of successful coevolution where human and plant populations change roughly at the same time.** Vertical dashed lines mark the timing of change for humans (cyan) and plants (pink). This parameter setting was taken as the default in the R implementation of the model.

distribution of types, from one centred in type 1 to one centred in type n (in Fig 5, a rightward movement in the top-right plots and upwards in the coevolution curves at the bottom left).

The coevolutionary trajectories can be divided into two phases:

- *Prior to* coevolutionary shift: This is a period during which human and plant populations are effectively coevolving. During this phase, population levels approach their first plateau or

pseudo-stable state value before coevolution takes effect while the distribution of types change—first slowly, then abruptly—towards the most mutualistic type. It ends when the change in the distribution of types can be considered completed in both populations; we define this moment to be the latest time step between timing$_H$ and timing$_P$ (in Fig 5, it is timing$_P$, represented by the pink vertical dashed line).

- *Following* coevolutionary shift: This is a period characterized by the stabilization of the populations around the truly-stable state. During this phase, both populations can be considered "changed" or effectively coevolved, even though they have still not realised the full potential for population growth made possible by coevolution. Although, depending on the specific conditions set by parameters, this phase typically involves a 'boom' for one or both populations.

Under some conditions, coevolutionary trajectories can display a punctual decrease in carrying capacities towards the end of the first phase, during the change from the least to the most mutualistic types. These demographic "bumps" happen in a population when the stronger mutualism type is less capable of exploiting other resources than the least mutualistic type (e.g., if $U_{bH1} > U_{bHn}$, then $U_{bH}[t] > U_{bH}[t + 1]$ during coevolution), while the other population has still not grown enough to counterbalance the loss in carrying capacity. In the example given in Fig 5, the plant population is the one suffering this effect, starting at the vicinity of the shift of the human population (vertical dashed cyan lines). In this case, the most mutualistic plant type is far less capable of exploiting non-anthropic resources than the least mutualistic type ($U_{bP1}$ or `utility_other_to_type_1_plants` = 100, $U_{bPn}$ or `utility_other_to_type_n_plants` = 20) and the utility given by the human population at that point ($U_{HP} \approx 80$) lies below the utility obtained from other resources when the least mutualistic types were the vast majority ($U_{bP}[t] \approx 100$, for t or `time` from 1 to 200).

**Coevolution occurs partially.**  Simulation experiments revealed cases in which the coevolution towards stronger mutualism occurs only partially. These cases are relatively rare, considering the entirety of the parameter space explored. However, they illustrate the complexity of the interaction of some factors accounted for in the HPC model.

The two types of end-states that fall into this general category, stationary suboptimal mutualism and oscillatory coevolution, are produced under parameter configurations that generally contain strong asymmetries either between the population or between types within the same population. These asymmetries include, for instance, configurations where one population has the most mutualistic types contributing the same amount of utility per capita than the least mutualistic types. In this scenario, the positive feedback between population growth and change in the distribution of types is weakened, but only enough to impede the change in one population; this is the case of the settings shown in Fig 6 ($\bar{U}_{H_1 P}$ or `utility_per_capita_type_1_humans_to_plants` = 0.5 and $\bar{U}_{H_n P}$ or `utility_per_capita_type_n_humans_to_plants` = 0.5).

## Parameter explorations

The extensive exploration of parameters demonstrated that a multiplicity of factors are involved in plant domestication and the origins of agriculture. However, the results also shed light on the relative importance of each of the factors included in the model.

We summarise the roles of the parameters of the model as 'facilitators', 'obstructors', and 'scalers' (Table 6). Under most conditions, increasing the values of any facilitator improves the chances of having a successful coevolution process, while greater values for obstructors will diminish it (respectively, positive and negative correlations with coevolution and dependency

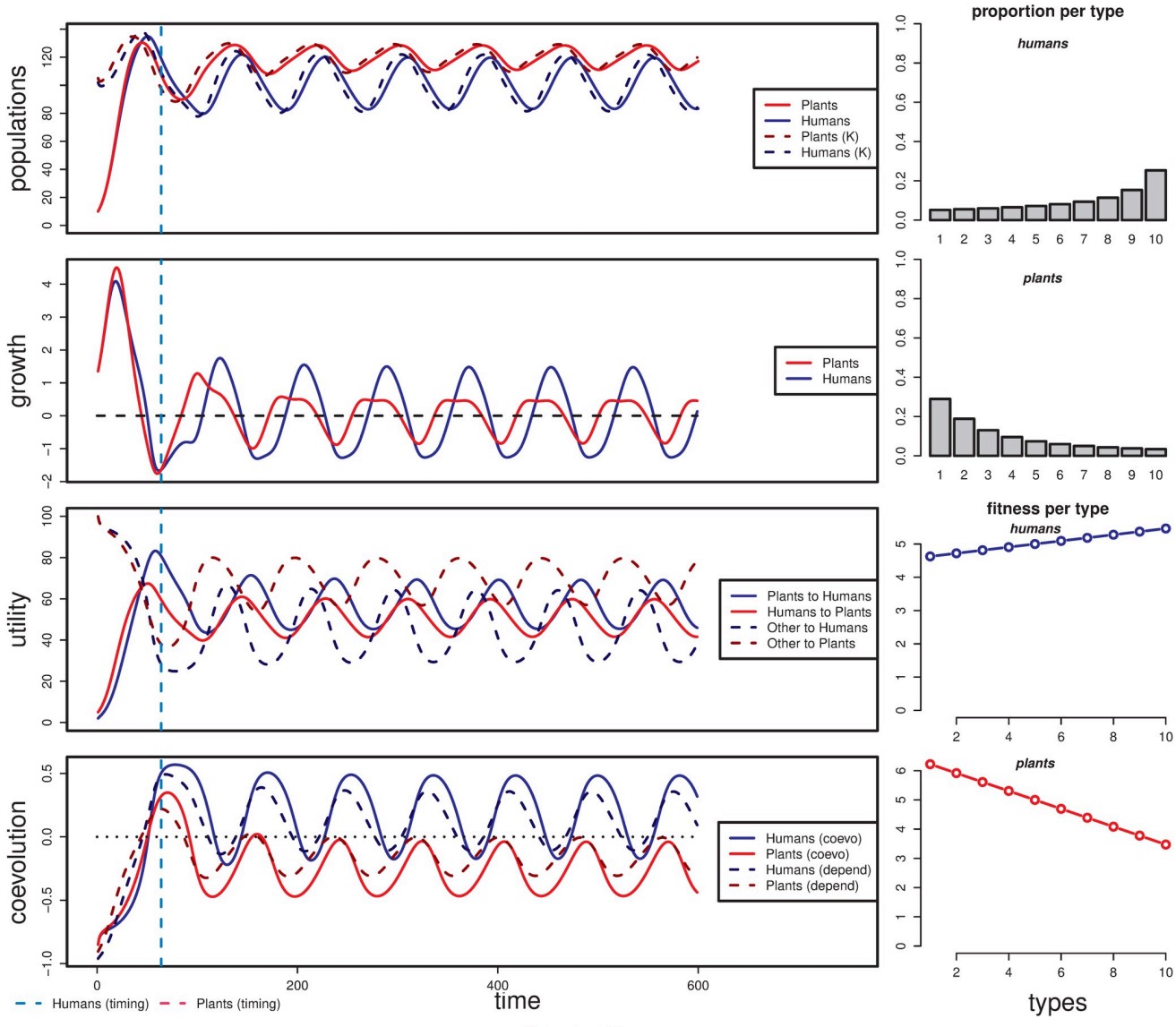

**Fig 6. Example of a simulation run with a case of partial oscillatory coevolution where only the human population fully transits to a majority of stronger mutualism types.** The timing of this change is marked by the vertical cyan dashed line.

coefficients). Scalers vary the size of populations (H or `humans`, and P or `plants`) at the end-state and the duration of the processes (time$_{end}$ or `time_end`, timing$_H$ or `timing_humans`, and timing$_P$ or `timing_plants`). Some parameters fit in more than one of the above classes, depending on the setting of the other parameters. The initial populations (ini$_H$ or `initial_population_humans`, ini$_P$ or `initial_population_plants`) remain outside this classification, having virtually no effect on end-states.

**Table 6. Parameter classification.**

| R notation | Math. notation | Facilitator | Obstructor | Scaler |
|---|---|---|---|---|
| `initial_population_humans, initial_population_plants` | $ini_H$, $ini_P$ | | | |
| `number_types_humans, number_types_plants` | $n_H$, $n_P$ | X | | |
| `undirected_variation_humans, undirected_variation_plants` | $v_H$, $v_P$ | X | | |
| `intrinsic_growth_rate_humans, intrinsic_growth_rate_plants` | $r_H$, $r_P$ | | | X |
| `utility_per_capita_type_n_plants_to_humans` | $\bar{U}_{P_nH}$ | X | | X |
| `utility_per_capita_type_1_plants_to_humans` | $\bar{U}_{P_1H}$ | X | | X |
| `utility_per_capita_type_n_humans_to_plants` | $\bar{U}_{H_nP}$ | X | | X |
| `utility_per_capita_type_1_humans_to_plants` | $\bar{U}_{H_1P}$ | X | | X |
| `utility_other_to_type_n_plants` | $U_{bPn}$ | (few cases) | X | X |
| `utility_other_to_type_1_plants` | $U_{bP1}$ | (few cases) | X | X |
| `utility_other_to_type_n_humans` | $U_{bHn}$ | | X | X |
| `utility_other_to_type_1_humans` | $U_{bH1}$ | | X | X |
| `max_area` | MaxArea | X | | X |

Within the range of values explored, all parameters but the initial populations and the intrinsic growth rates ($r_H$ or `intrinsic_growth_humans`, $r_P$ or `intrinsic_growth_plants`) displayed tipping points, i.e. threshold values beyond which the end-states of simulations change drastically (non-linear effect). The exact location of a tipping point in one parameter depends on the values of all others parameters with tipping points, indicating a generally strong interaction between their effects, and hence no single-cause explanation for a given end-state can be accurate. For instance, in light of the trajectories in Figs 4 and 5, it would be correct to say that coevolution occurs in the latter *because* `utility_per_capita_type_1_plants` is high enough (i.e. higher than a threshold value between 0 and 0.15). Yet, it is also correct to affirm that it is so because, simultaneously, `utility_other_to_type_n_humans` is low enough, `undirected_variation_plants` is high enough, and so on.

Despite their shared explanatory role, parameters vary significantly in importance when predicting the values of the coevolution coefficients at the end-state. We were able to rank the explanatory power of each parameter by fitting Random Forest Regression models where parameters are inputted as predictors in respect to each coevolution coefficient separately (Fig 7).

The same procedure was applied for the dependency coefficients and timings; see section 5.2 in [68]. The assessment of parameter importance for the dependency coefficients displayed a similar pattern, only highlighting those parameters with a direct impact on the carrying capacity of the respective population (greens and blues). While the intrinsic growth rates have the highest impact on the timing of coevolution, all other parameters are scored similarly, having at least some importance for one or both populations. Parameter explorations revealed that timing indicators (timing$_H$ or `timing_humans`, timing$_P$ or `timing_plants`, and $t_{end}$ or `time_end`) are larger, the closer parameter values are to a tipping point. In those liminal cases, the coevolutionary process can take up to three times longer.

**Number of types, undirected variation and intrinsic growth rate.**   The numbers of types in human and plant populations ($n_H$ or `number_types_humans`, $n_P$ or `number_types_plants`) facilitate change (i.e. facilitators). However, these two parameters stand out as the least important. Such a result is desirable given that the aspect regulated by these parameters—i.e. the discretionality of population variation—is a necessary artefact of the model and

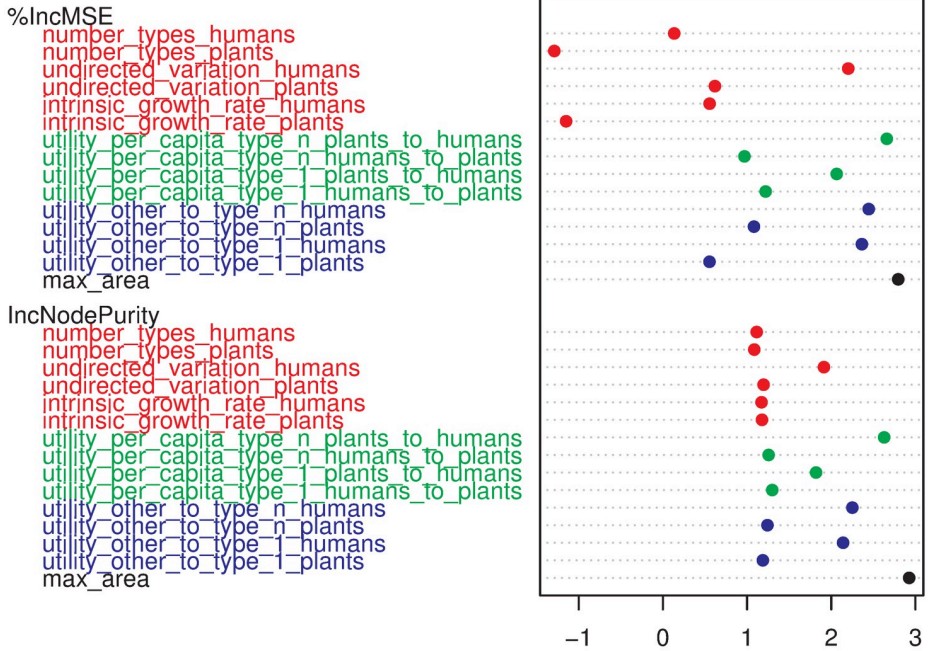

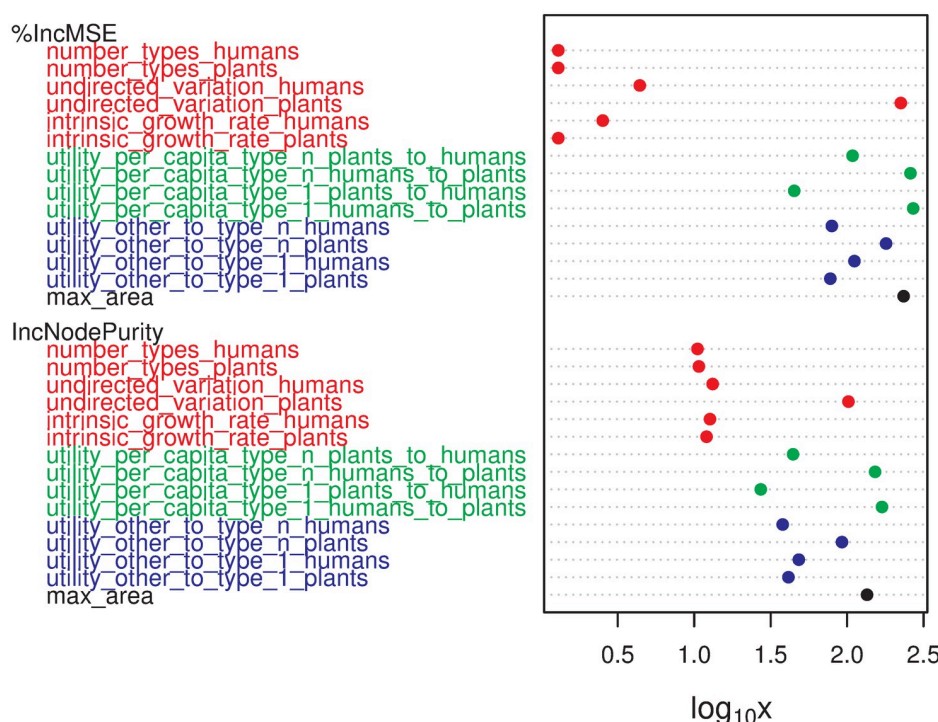

**Fig 7. The importance of parameters measured as a percentage of mean squared error increase (%IncMSE) and total decrease in node impurities (IncNodePurity) obtained by fitting Random Forest Regression models where parameters are inputted as predictors of the human (left) and plant (right) coevolution coefficients; for similar applications, see [69, 70].** The number of trees and number of sampled variables were optimized by a standard 10-fold cross-validation procedure [71].

can only translate to arbitrary classifications when regarding real populations. Ultimately, every individual in a real population could be a single instance of their own type. The overall low importance of these parameters warrants future explorations to treat these as constants, preferably setting them at values much greater than unity ($n_A \gg 1$).

The levels of undirected variation ($v_H$ or `undirected_variation_humans`, $v_P$ or `undirected_variation_plants`) are also facilitators. With higher variation, there are more individuals belonging to stronger mutualism types. Though unfit for the initial conditions, these are the pioneer individuals that may eventually build up the necessary selective pressure on the partner population and trigger coevolution. The positive relationship between undirected variation and the occurrence of coevolution agrees with Fisher's fundamental theorem of natural selection [72, 73], according to which higher variance increases the rate of adaptation of a species; which, in this case, leads to stronger mutualism.

Intrinsic growth rates ($r_H$ or `intrinsic_growth_rate_humans`, $r_P$ or `intrinsic_growth_rate_plants`) are scalers, conditioning how fast populations levels change. Generally, higher intrinsic growth rates return shorter periods of population growth and change of type distribution. However, because they also define how rapid is the feedback cycle regulating the mutualistic selective pressures, they show a mirrored pattern where the intrinsic growth rate of one population has its greatest impact on the timing of change of the other population.

**Utility-related parameters.**   Overall, the most important parameters in the HPC model are those characterising the potential of the mutualistic interaction between humans and plants (Fig 7); i.e. the utility per capita of type n individuals to the other population ($\bar{U}_{P_nH}$ or `utility_per_capita_type_n_plants_to_humans`, $\bar{U}_{H_nP}$ or `utility_per_capita_type_n_humans_to_plants`). Although the correspondent values for type 1 individuals ($\bar{U}_{P_1H}$ or `utility_per_capita_type_1_plants_to_humans`, $\bar{U}_{H_1P}$ or `utility_per_capita_type_1_humans_to_plants`) also play a significant role, coevolution is more often enabled by the utility given by the higher-end types in the mutualistic spectrum. The effect of these parameters is mirrored (greens in Fig 7): `utility_per_capita_type_n_plants_to_humans` mostly affects change in the human population, and `utility_per_capita_type_n_humans_to_plants` does it in the plant population. However, `utility_per_capita_type_n_plants_to_humans` weights considerably on both humans and plants.

All four parameters related to the utility exchange between humans and plants set a range of utility per capita of each population type that amounts to population totals (e.g., $U_{HP}$ or `utility_humans_to_plants`). Whenever these totals overcome the totals given by the other resources (e.g., $U_{bP}$ or `utility_other_to_plants`), the fitness scores will favour stronger mutualism types and trajectories will shift towards a successful coevolution (Fig 8).

The parameters determining the utility given by other resources ($U_{bH1}$, $U_{bHn}$, $U_{bP1}$, and $U_{bPn}$) are obstructors. Overall, the parameters corresponding to the human population ($U_{bH1}$, $U_{bHn}$) have a stronger effect than those related to plants (blues in Fig 7). The two parameters regulating the utility of other resources to plants ($U_{bP1}$, $U_{bPn}$) can also be facilitators depending on the conditions set by other parameters; however, their effect is the weakest of all eight parameters associated with utility (greens and blues in Fig 7).

The parameters associated with utility are also important scalers since they have a direct effect on carrying capacities. The parameters contributing to the carrying capacity for humans ($\bar{U}_{P_1H}$, $\bar{U}_{P_nH}$, $U_{bH1}$, and $U_{bHn}$) are able to influence scale more freely because they are not capped by MaxArea. In particular, the utility of other resources to type 1 individuals ($U_{bP1}$, $U_{bH1}$) can condition almost entirely the respective carrying capacity—and consequently the

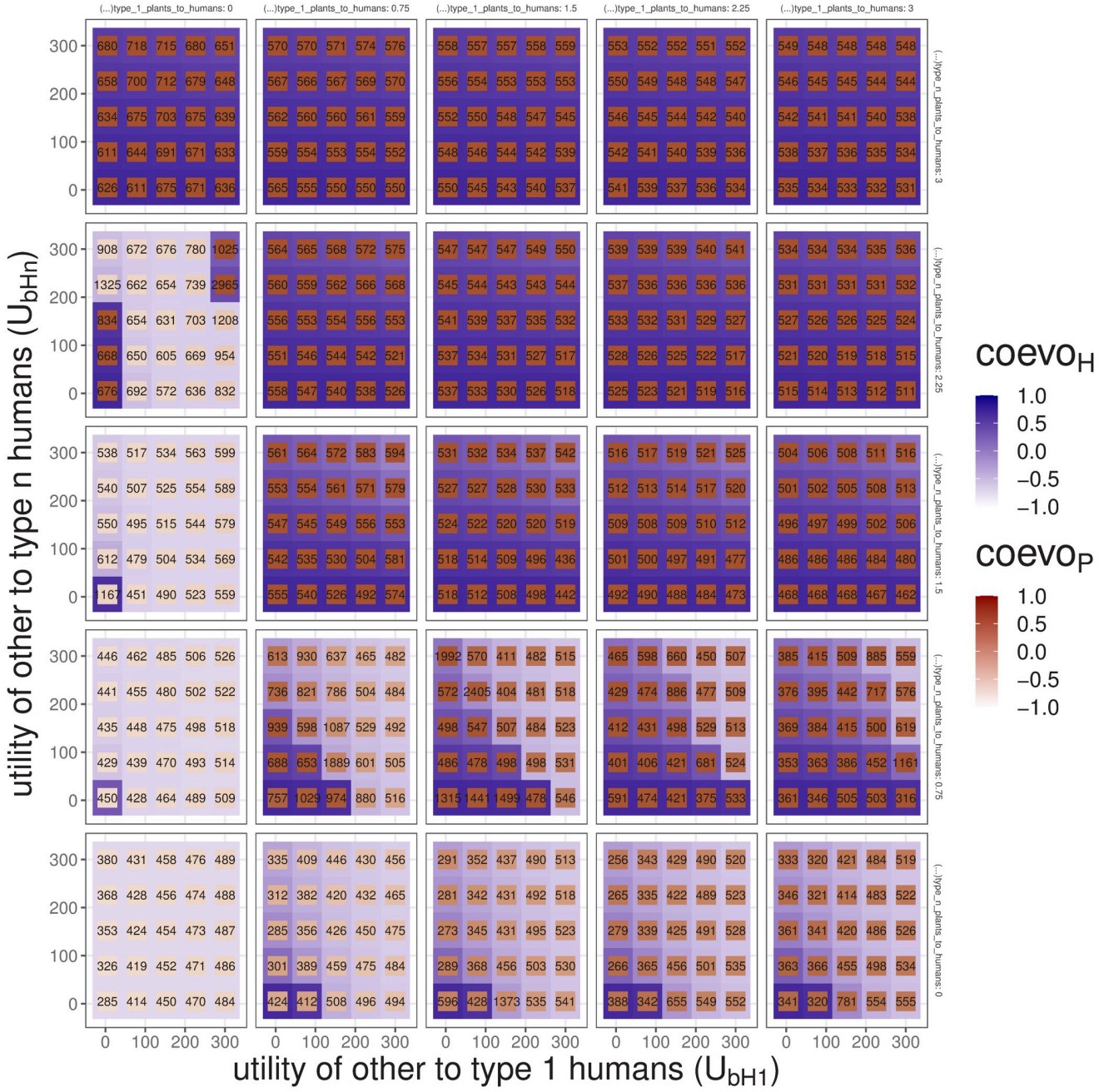

**Fig 8. The coevolution coefficients and $t_{end}$ resulting from a four-parameter exploration of $\bar{U}_{P_1H}$, $\bar{U}_{P_nH}$, $U_{bH1}$, and $U_{bHn}$.** The plot depicts examples of facilitators ($\bar{U}_{P_1H}$ and $\bar{U}_{P_nH}$; values of large grid), and obstructors ($U_{bH1}$ and $U_{bHn}$; values of small grids).

population levels—at the end state. These parameters alone can generate trajectories where the human population at the end-state varies from a few to thousands of individuals, without ever incurring in coevolution.

Trajectories with coevolution can be very different (compare Figs 3E to 5) mainly due to the amount of space available for plants (MaxArea) and the conditions regulating the mutual utility between humans and plants ($\bar{U}_{H_1P}$, $\bar{U}_{H_nP}$, $\bar{U}_{P_1H}$, and $\bar{U}_{P_nH}$). These are important

facilitators, but also have the potential for producing end-states that differ dramatically in the sheer size of the human and plant populations (H, P). For instance, an overall low utility of plant types to humans ($\bar{U}_{P_1 H}$, $\bar{U}_{P_n H}$) can still produce end-states with coevolution that are indistinguishable in terms of human population size from others without coevolution, where the overall utility of other resources to humans is sufficiently high.

Surprisingly, full-fledged coevolution can still happen when type n individuals contribute less than type 1 individuals (e.g., $\bar{U}_{P_n H} < \bar{U}_{P_1 H}$). For instance, when $\bar{U}_{P_n H} = 1.5$ and $\bar{U}_{P_1 H} = 3$ in Fig 8. This happens whenever the population total (e.g., $U_{PH}$) overcomes the amount given by other resources (e.g., $U_{bH}$). This discovery indicates that, at least under the assumptions of this model, the adaptation to mutualism could cause the deterioration of the contribution of individual organisms while still increasing population numbers.

## Discussion

Much of the groundwork that helped to understand the evolutionary dynamics of plant domestication comes from archaeology, and more specifically from archaeobotany. Harris [74] theorised the process of domestication as composed of three stages: 1) wild food procurement by hunting and gathering societies; 2) cultivation of wild plants; and the 3) domestication syndrome fixation that established true agriculture of domestic plants. The early plant datasets, mostly coming from the Fertile Crescent, were interpreted as suggesting a 'rapid transition' between these stages due to a strong and direct human selection favouring interesting characters, such as non-brittle spikelets in cereals [75] and suppression of seed dormancy in legumes [76]. However, the richer archaeological record of the last few decades suggests that such transitions could involve a period of pre-domestication cultivation lasting thousands of years [77, 78], followed by fixation of the emerging domestic traits, a process that again can happen over thousands of years; see e.g. for cereals [79]. This mechanism, leading to the evolution of domesticated and commensal species, seems to have been a response to the emergence of human-modified environments appearing from the end of the last glaciation [80]. Both the domesticated plants and human populations benefited from this co-evolutionary process, leading to stronger mutualism [53].

### Multiple factors, multiple scenarios

The HPC model illustrates the multiplicity of the dynamics that, under its theoretical framework (Tables 4 and 5), can explain ecological and socio-economic shifts, such as the so-called neolithisation. The exploration of the model reinforces the premise that, to explain the domestication of plants and the adoption of agricultural practices, we must integrate the different degrees of complexity of the phenomenon itself, and accept that single-factor explanations do not fit this multiple and heterogeneous reality [1, 5]. The great variety of scenarios regarding the characteristics of the crops and of the ecological milieus, as well as the different social, cultural and technological settings in human populations, highlights the complexity of the process and the inevitability of generating case-specific narratives when interpreting the evidence. However, the HPC model goes beyond the replication of multiple single-case idiosyncrasies and contains the formalisation of a general mechanism: the coevolution of humans and plants. This model is able to generate a wide diversity of simulated trajectories and end-states, expressed as aggregated quantitative variables, which we hope can be used in the future to produce explanatory frameworks for specific real-world cases. Therefore, the HPC model is not aimed at reproducing historical processes *per se* but different possible scenarios of human-plant coevolution, which can be searched in and contrasted with specific lines of evidence.

The model points to several aspects that can explain the emergence of agricultural systems. Some of these aspects, like the utility *per capita* to the other population, have been part of the archaeological and botanical discourse, albeit not as a formal model [75]. Furthermore, the model shows that a small increase or decrease around a threshold value can produce major changes in the system (tipping point) and that, for coevolution to occur, all parameters showing tipping points must be either beyond or below a particular threshold, which, in turn, depends on the values of all other parameters.

The HPC model also shows that certain differences between human and plant populations can have an important effect on the outcome of human-plant coevolution. The selective pressure of one versus the other may vary significantly among parameter settings, thus producing qualitatively different scenarios.

At one end of the mutualism spectrum, the model can generate scenarios where the subsistence relies heavily on the plant population and the selective pressure is sufficient to drive a substantial change on plant type frequency and population levels, thus leading to some form of agricultural system. At the other end, the model produces outcomes where there is low human-on-plant pressure and humans have many (and preferred) alternative food sources. In such instances, wild plant forms are maintained in the population and low densities are retained. Human subsistence in such cases relies mostly upon other resources, which might still allow for high population densities independently of the plant population; e.g., fishing and complex hunter-gatherers [81, 82]. Between these extreme end-state scenarios, the model also simulates other "realities" in which only one population exerts enough selective pressure over the other for it to shift towards stronger mutualism types: societies cultivating plants that, though affected, remain not fully domesticated (cultivation without domestication), or those foraging plant populations that increase their productivity without humans investing more time in them (domestication without cultivation).

## Intensification and the coevolutionary dynamics of prehistoric plant management

In most early cases, the adoption of agriculture seems to be the culmination of a long process with deep roots in hunter-gatherer societies [83]. Archaeological literature traditionally considers this process to be fuelled by a series of changes related to food resource diversification [84, 85] and, particularly for plants, intensification [86–89]. Within this context of change, intensive gathering and cultivation have been considered economic practices within a continuum, where some plant species are gathered opportunistically and others systematically exploited. At the beginning of every transition to agriculture, predatory strategies (fishing, hunting, and gathering) were central to human subsistence, while mutualism (plant tending and animal husbandry), if any, were complementary [32].

The theoretical continuum between resource management, domestication, and agriculture assumes that the existence of each forgoer component is paramount for the development of the next "step". However, any one of these phenomena does not inevitably lead to the next [4]. Assuming that in some cases there is an effective transition to agriculture, that means that the focus shifts from a wide range of prey-like resource use to a relatively small number of very successful mutualism partners, among which domesticated plants eventually become the basic source of staple food. In this framework, the coevolution between humans and plants can be defined as a process mediating between weaker and stronger mutualism that can involve many stages, each with a qualitative change in the distribution of types and consecutive boom and stabilisation of both populations.

The HPC model allows identifying various regimes of mutualism between humans and plants. The model, in fact, can give rise to a wide range of scenarios that, from the human point of view, consist of different combinations of wild/domesticated plant food resources and modes of exploitation of such resources, with variable commitments in terms of diet and investment. These strategies can be interpreted as mixed economies, which have been shown to be possible, viable and even resilient socio-economic choices. Within the specialized literature, mixed economies are usually understood as minor or marginal socio-economic systems, defined either as the combination of different strategies of low-level food production [90] or as by-products of a transitory, and thus not stable, stage [91].

These strategies are not necessarily implemented as static combinations, but also as seasonal or periodical activities, shifting from one strategy to another [92, 93]. In addition, the pursuing of such strategies might not be a clear and rational decision adopted by specific social agents or groups in charge of the economic activities, but a scenario arising by the aggregation of multiple decision-making processes at the community level, throughout generations.

There is a strong relationship between richness of viable economical options and the specialisation and diversification in subsistence strategies [94–97]. Specialisation and diversification are hypothesised to have first occurred during the Mesolithic as a mean to intensify the acquisition of resources and they are considered a preamble for the implementation of agricultural practices [98]. Although the concept of intensification could support the continuum concept, there is a strong debate about the reasons and conditions under which intensification takes place in hunter-gatherer societies [99].

With the current work, we aim at showing how the succession of mixed economies are an intrinsic part of the coevolutionary dynamics between human and plants, and shed some light on why can these culminate, in many cases, in the emergence of agriculture.

## Insights on the Neolithic Demographic Transition

In archaeological theory, the origins of agriculture is often defined as the birth of a new socio-economic paradigm involving key changes in human demography and social organization, such as increased hierarchy and division of labour. Among these changes, the most striking is the unprecedented population growth that usually followed the adoption of agriculture, i.e. the Neolithic Demographic Transition [100–102].

The HPC model considers the relationship between plant utility and human needs (population pressure) but also the positive effects humans can have on plant growth. The latter involves a delayed improvement of plant utility to humans, through the evolution of traits and sheer population growth, and an increasing human population growth, putting pressure on old and new food resources. Low population pressure, given by either low population density or abundance of food resources, has been argued as a precondition for increasing growth rates in human populations [93]. The demographic increase by the end of the Upper Palaeolithic, as shown by the archaeological record, has been considered a possible cause for a series of intensification processes (such as the intensification of plant gathering or the expansion in coastal populations and an increase in the consumption of coast and marine resources). At the same time, either the intensification of resource exploitation and/or the adoption of agricultural practices (both increasing the productivity per area but also involving labour-intensive, time-sensitive activities) might have fostered the abandonment of a series of measures controlling fertility, resulting in a population increase.

A few studies have recently focused on the various demographic booms and busts identified during the Early Neolithic in Europe [16] and which may be interpreted as the possible diverse outcomes of the neolithisation process. While neolithisation intuitively implies a population

boom due to the overall increase in food availability, not all instances of shifting to an agricultural economy appear to have been demographically successful. The HPC model suggests a possible explanation for population busts within its formal framework: a momentary decrease in the adaptive fitness of the population and, thus, of the carrying capacity of the environment.

The growth of the human population can have a series of implications. First, a higher demand of the available resources that become manifest in the selective pressure on the plant population or other resources (mixed economy). This may have positively affected the domestication process, by increasing plant bulk productivity, but also produced a series of changes fostering the hunter-gatherer strategy to be less effective when combined with a more invested plant cultivation. When cultivation becomes a priority, there is an expectation for societies or groups within societies to become more sedentary, at least seasonally, so that crops are properly monitored during growth. As a consequence, there would be a reduction in the fitness of the hunter-gatherer strategies. Firstly, because some expertise may be lost, even within a generation, as a considerable part of the labour and efforts for cultural transmission would be focused on cultivation. Secondly, with sedentism (or partial sedentism), the catchment area available for foraging would shrink and quickly be impoverished, having less time to recover and at the same time suffering the effects of expanding cultivation practices. Thirdly, the human population will be pressured to adapt to the needs and schedule of the cultivated plant species and the associated labour bottlenecks, which might be incompatible with the dynamics required for gathering or hunting specific wild resources.

## Conclusions

Considering the potential of the modeling results, we would like to underline the bullet and conservative nature of the HPC model. All the diversity observed in terms of both attractors and trajectories was generated by the combination of only two submodels, the Verhulst-Pearl Logistic equation and the Replicator Dynamics, which are straightforward benchmark models in theoretical biology. The sole fact that a relatively bullet model can greatly help to understand complex phenomena, such as the origin of agriculture, argues for the use of formal models, and specifically for simulation approaches, in archaeology.

As other examples in the past [53, 55], the HPC model demonstrates that population-level (top-down) theory can still produce useful insights. Strong explanatory frameworks can be achieved without the fine insights of case-wise detail; an approach often resisted by archaeologists, but which is at the same time accepted whenever data is interpreted. In this sense, we consider that formal models are fundamental tools to present, demonstrate and explore any theoretical proposal. The HPC model also offers a solid basis for the design and further development of generative (bottom-up) models [51, 52, 103–105], and is complementary to approaches focusing on plant domestication syndrome through phenotypic and genetic characterisation [106, 107].

According to the HPC model, there are several factors involved in the facilitation or obstruction of the emergence of agricultural systems. Although the model confirms the expectation of attributing several causes to the origin of agriculture, it also further explains how multiple factors could be compatible with asserting causation in a historical sense (i.e., concatenation of events).

In the HPC model, the state of the system connecting humans and the plant species is sensitive to almost the totality of the thirteen parameters. More precisely, this sensitivity is expressed as a rather abrupt shift (tipping point) from a weak to a strong mutualistic state, or vice-versa, depending on the threshold values for each parameter, which are in turn dependent on the current values of every other parameter. Then, according to our model, the emergence

of agriculture could be explained by the confluence of all these conditions at specific times and places. However, it seems unlikely that, for the same case of emergence, all these conditions change and cross multiple thresholds simultaneously. Conversely, still within the HPC model, we may envisage scenarios in specific regions at specific moments (i.e. under a specific set of other conditions) where the change in few or even one condition triggered the emergence of agriculture. In this case, certain factors may be considered the cause of the phenomenon in a more deterministic sense.

Beyond the identification of factors that play a role in the human-plant coevolutionary dynamics, the HPC model allows assessing the differences in scale and timing between case trajectories. This capability seems to be especially relevant to understand the many cases of non-industrial agricultural systems documented by archaeology and ethnography. By controlling parameter on a case-by-case basis, further work with the HPC model would yield insight on the reliability of particular hypotheses of how agricultural systems emerged in the past, and help explaining why some origins are more observable than others.

## Acknowledgments

We would like to thank the reviewers, particularly Isaac I. Ullah, for reading the manuscript and source code and providing good and insightful comments. Fig 2 contains modified versions of the Plant icon release by Bakunetsu Kaito under CC-BY 3.0 (available at: https://thenounproject.com/icon/961532/) and the People icon released by Fahmihorizon under CC-BY 3.0 (https://thenounproject.com/icon/1426355/).

## Author Contributions

**Conceptualization:** Andreas Angourakis, Marco Madella, Debora Zurro.

**Data curation:** Andreas Angourakis.

**Formal analysis:** Andreas Angourakis, Jonas Alcaina-Mateos.

**Funding acquisition:** Andreas Angourakis.

**Methodology:** Andreas Angourakis, Jonas Alcaina-Mateos.

**Software:** Andreas Angourakis, Jonas Alcaina-Mateos.

**Visualization:** Andreas Angourakis.

**Writing – original draft:** Andreas Angourakis, Marco Madella, Debora Zurro.

**Writing – review & editing:** Andreas Angourakis, Jonas Alcaina-Mateos, Marco Madella, Debora Zurro.

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
