## [Decision Letter · Decision Letter 0]

11 Apr 2022

PONE-D-21-36515

Human-Plant Coevolution: A modelling framework for theory-building on the origins of agriculture

PLOS ONE

Dear Dr. Angourakis,

Thank you for submitting your manuscript to PLOS ONE. After careful consideration, we feel that it has merit but does not fully meet PLOS ONE’s publication criteria as it currently stands. Therefore, we invite you to submit a revised version of the manuscript that addresses the points raised during the review process.

As you address the reviewers' comments, we would like for you to attend in particular to:  (1) Reviewer #1's requests for clarification of the text, including their concerns that (a) the model is not currently as well linked to general theory as it could be and (b) the manuscript's treatment of neolithization as discrete and stage-like may have negative implications for interpretation of the model's results; (2) Reviewer #2's request that you rename variables in both the code and related text according to "clean code" best practices, versions of which are available on-line; and (3) proper numbering of figures throughout the manuscript.

We look forward to receiving your revised manuscript.

Kind regards,

Raven Garvey, Ph.D.

Academic Editor

PLOS ONE

Journal Requirements:

"This research is part of the activities of the Culture and Socioecological Dynamics 

Research Group (CaSEs), a Quality Group of the Generalitat de Catalunya (2017 

SGR212) (JAM, MM, DZ), and was supported by the CULM project 

(HAR2016-77672-P), funded by the former Spanish Ministry of Economy and 

Competitiveness (MINECO) (DZ), and the TwoRains project, funded by the European 

Research Council (ERC) under the European Union’s Horizon 2020 research and 

innovation programme (grant agreement no 648609) (AA). Previous development was 

also made possible through the support of the SimulPast Project— Consolider Ingenio 

2010 (CSD2010-00034), funded by the former Spanish Ministry for Science and 

Innovation (MICIN) (AA, JAM, MM, DZ), and the CAMOTECCER project 

(HAR2012-32653) and FPI contract (BES-2013-062691), funded by MINECO (AA).

Figure 2 contains modified versions of the Plant icon release by Bakunetsu Kaito under CC-BY 3.0 (available at: https://thenounproject.com/icon/961532/) and the People icon released by Fahmihorizon under CC-BY 3.(https://thenounproject.com/icon/1426355/)."

"This research is part of the activities of the Culture and Socioecological Dynamics Research Group (CaSEs), a Quality Group of the Generalitat de Catalunya (2017 SGR212) (JAM, MM, DZ), and was supported by the CULM project (HAR2016-77672-P), funded by the former Spanish Ministry of Economy and Competitiveness (MINECO) (DZ), and the TwoRains project, funded by the European Research Council (ERC) under the European Union's Horizon 2020 research and innovation programme (grant agreement no 648609) (AA). Previous development was also made possible through the support of the SimulPast Project— Consolider Ingenio 2010 (CSD2010-00034), funded by the former Spanish Ministry for Science and Innovation (MICIN) (AA, JAM, MM, DZ), and the CAMOTECCER project (HAR2012-32653) and FPI contract (BES-2013-062691), funded by MINECO (AA)."

Reviewers' comments:

Reviewer's Responses to Questions

**Comments to the Author**

1. Is the manuscript technically sound, and do the data support the conclusions?

Reviewer #1: Yes

Reviewer #2: Yes

Reviewer #3: Partly

2. Has the statistical analysis been performed appropriately and rigorously? 

Reviewer #1: Yes

Reviewer #2: Yes

Reviewer #3: I Don't Know

3. Have the authors made all data underlying the findings in their manuscript fully available?

Reviewer #1: Yes

Reviewer #2: Yes

Reviewer #3: Yes

4. Is the manuscript presented in an intelligible fashion and written in standard English?

Reviewer #1: Yes

Reviewer #2: Yes

Reviewer #3: Yes

5. Review Comments to the Author

Reviewer #1: In this manuscript, Angourakis and colleagues develop a fairly complex and abstract mathematical model for the human-plant coevolution that was ostensibly necessary for the wholescale adoption of agriculture. In general, I found the manuscript well-written, the logic of the model well laid-out and the implications of the model important, mainly in that they result in quite a few hypotheses that ought to be testable with archaeological and ecological data. This, I think, is the main contribution of the reported research. I like that the model they develop instrumentalizes the potential of earlier systems theory, taking advantage of modern computing’s power to develop and run complex, multiiterative analyses. I will say that though I understood the logic of the model and thought it was well explained, I did not delve deeply into the actual formulae to make sure they all worked as claimed, so I can’t comment on that end of this analysis. In at least two places, the authors seem to imply that the process of domestication through co-evolution was indeed as complex as modeled (e.g., lines 458-459), though of course that remains to be seen: it depends on how well the archaeological and paleoecological data conform to the expectations of the model Angourakis and colleagues developed, something they eschew in this manuscript. It’s entirely possible that domestication processes were simpler than they model. I think this fact should be made clear.

Though I see a lot of value in this research, I do think the authors could improve and clarify a few things. First, they seem to imply they are theory building, which they are not. What they do is develop a model derived from general theories of gene-culture co-evolution and, perhaps to some degree, niche construction theory (NCT). To this end, I think the connections of their model to general theory could be made more explicit, particularly in the background/literature review. Research by Pete Richerson, Robert Boyd, Bruce Winterhalder, Melinda Zeder, and the like is relevant in this regard. In this context, it’s worth looking at the sometimes vehement debate between NCT theorists like Zeder and more behavioral-ecology (HBE) minded theorists like Piperno (both of whom are cited in the MS) and whether or not they are really all that alike, or if they are different (see lines 28-33). In short: though I see opportunity to marry these two perspectives, most researchers see them as opposed, in part due to the way they see agency playing out (see line 34). In fact, I see Angourakis and colleagues’ model articulating quite well with both HBE and niche construction advocates (worth looking at chapters in Kennett and Winterhalder’s “Behavioral Ecology and the Transition to Agriculture” here).

Second, particularly in the discussion section, but also in the paper more generally, the authors treat “Neolithisization” as a discrete, stage-like event, a result, at least in part, of a cursory presentation of non-agricultural (i.e., hunter-gatherer) behavioral diversity. A lot of current thinking on hunter-gatherers and their relationship to agriculturalists is that most or all of the behaviors necessary for agriculture—generation of surplus, storage, landscape management, concepts of private property, etc. are found in the diversity of hunter-gatherer behavior. I’d encourage the authors to look more closely at this literature and reframe their general approach and discussion to recognize that most of the behaviors we associate with agriculturalists, with the exception of domesticated plants, were likely present in many nonagricultural societies well before domestication. This could change their interpretation of their model’s results as well

One note, the mention of bioarchaeology in the abstract seems odd, as the authors themselves attest to the wide range of different studies outside of bioarchaeology that attest to the chronology of domestication and the development of agriculture.

On a final note, it was hard to tell which figures were being referred to in the text due to an absence of figure numbers (and corresponding figure captions). I think I eventually figured it out, but some good figure captions would really help the reader connect the text to the images.

Reviewer #2: The model presented here is useful and makes a good contribution to pushing forward theory on the mutualistic pathway of plant domestication. I commend the efforts to make the code available and the creation of a relatively user-friendly shiny app! However, a major issue with the code and how it is decribed in the text is the use of non human-readable variable names throughout. This greatly complicates the ability to read, understand, and reuse the code, and makes it very difficult to follow the narrative of the model logic in the paper. I strongly, strongly recommend that these variable be renamed using the accepted readable variable naming conventions that are common in open/reproducible code (e.g. CamelCase or underscore_case with short but descriptive variable names such as "InitialHumanPopulation" or "plant_type_n_coevolution_coefficient"). Yes, the variable names are longer, but they are actually readable and make it much easier to understand (and thus critique) the code and the narrative of the model function. There also seemed to be an issue with the in-text figure numbering, which made it difficult to follow the results section. I have a few citations I'd like to see added as well, so I am attaching a marked up version of the manuscript with specific recommendations. I think the substantial conclusions and discussion are very good, with a few minor suggestions for how to extend these (again, in the attached marked up PDF). In general, I think this is a highly effective and important work, and deserves to be published with these minor revisions. Also, I am happy to chat about any of this and don't care to remain anonymous -- feel free to send me an email! iullah@sdsu.edu

Reviewer #3: First, it's not clear to me what this research was intended to achieve. The closest we get to this is in lines 545-7: "...we aim to show how the succession of mixed economies are intrinsic parts of coevolutionary dynamics between humans and plants, and illuminate why these culminated, in many cases, in the origins of agriculture." If this was indeed the aim then, despite claims to the contrary (see below), the paper failed to achieve this.

Rather than addressing the 'how' and 'why' questions promised in the abstract and introduction, the paper is more descriptive than analytical, nor was it clear how the HPC model could be used in the future to address how and why questions concerning individual cases studies or in search of some common or overarching similarities in the trajectories to agriculture in different contexts.

I therefore found the conclusions drawn towards the end of the paper unsubstantiated by the research as presented. For example: the conclusions that the HPC model “can greatly help understand…the origins of agriculture” or “can produce useful insights” or “offers a solid basis for the development of generative models” – where is the evidence for this in the paper? How would the model achieve such objectives? Greater clarity of purpose and a more realistic account of achievement is needed.

6. PLOS authors have the option to publish the peer review history of their article (what does this mean?). If published, this will include your full peer review and any attached files.

Reviewer #1: No

Reviewer #2: **Yes: **Isaac Ullah

Reviewer #3: No

---

## [Author Response · Author response to Decision Letter 0]

7 Jul 2022

PONE-D-21-36515 – Human-Plant Coevolution: A modelling framework for theory-building on the origins of agriculture

Response to Reviewers

Authors: We would like to thank the reviewers for reading the manuscript and providing good and insightful comments. We have made our best to address them and improve the manuscript. All the changes in the manuscript have been written so that changes can be properly tracked.

Andreas Angourakis: As the submitting author, I personally apologise to editors and reviewers for the issue with the numbering of figures. It was caused by how I suppressed the LaTeX code that added the figures in the text, as PLOS ONE requires figures to be added separately. I failed to realise that this action broke the reference link that assigns figure numbers automatically, besides not printing the figure captions at the figure insert location, as it is specified in PLOS ONE LaTeX template. This error is now corrected.

Reviewer #1

In this manuscript, Angourakis and colleagues develop a fairly complex and abstract mathematical model for the human-plant coevolution that was ostensibly necessary for the wholescale adoption of agriculture. In general, I found the manuscript well-written, the logic of the model well laid-out and the implications of the model important, mainly in that they result in quite a few hypotheses that ought to be testable with archaeological and ecological data. This, I think, is the main contribution of the reported research. I like that the model they develop instrumentalizes the potential of earlier systems theory, taking advantage of modern computing’s power to develop and run complex, multiiterative analyses. I will say that though I understood the logic of the model and thought it was well explained, I did not delve deeply into the actual formulae to make sure they all worked as claimed, so I can’t comment on that end of this analysis. In at least two places, the authors seem to imply that the process of domestication through co-evolution was indeed as complex as modeled (e.g., lines 458-459), though of course that remains to be seen: it depends on how well the archaeological and paleoecological data conform to the expectations of the model Angourakis and colleagues developed, something they eschew in this manuscript. It’s entirely possible that domestication processes were simpler than they model. I think this fact should be made clear.

Authors’ reply:

We acknowledge the reviewer’s comments and concerns about any implicit assumptions in our statements. The model is proposed in the spirit of theory-building, so it is certainly not our intention to suggest that such a model is the only possible and valid model. We carefully checked the specific paragraph given as an example and revised the remaining text, having the reviewer’s suggestion in mind. We have changed the wording, and we hope that now our reasoning is more clearly expressed.

More concretely, we modified the fragment at the beginning of the subsection in Discussion, “Multiple factors, multiple scenarios”, starting in line 458 of the old manuscript:

>”The HPC model illustrates the multiplicity of the dynamics that, under its theoretical framework (Tables~\\ref{table:assumptEco}~and~~\\ref{table:assumptCoevo}), can explain ecological and socio-economic shifts, such as the so-called neolithisation. The exploration of the model reinforces the premise that, to explain the domestication of plants and the adoption of agricultural practices, we must integrate the different degrees of complexity of the phenomenon itself, and accept that single-factor explanations do not fit this multiple and heterogeneous reality~\\cite{Aiello2011a,Fuller2014}.”

To clarify, what we would like to suggest in the subsection ‘Multiple factors, multiple scenarios’ of the Discussion is that there is no single explanation that allows an understanding of the emergence of agriculture and plant domestication in the different areas of the globe where this phenomenon has been observed. Despite the number of specifications that the formalisation of the coevolution phenomenon required, we consider that the HPC model is in fact a rather simple version of what we know as coevolution (mechanism). For example, regarding the plant side, the model does not delve into the currently well-established complexity of genetics and epigenetics in plants. Additionally, if plant domestication had occurred through less symmetric coevolution (e.g., relying more heavily on human design and intentionality, as it is argued by the proposers of the “fast domestication”), the formal description of the model would actually have to be more complex, not more simple. Our argument here is that even a very basic description of coevolution already points to a considerable diversity of trajectories, i.e., the dynamics that we can then compare to empirical data). The model shows the multiplicity of possible scenarios, some are more simple while others are more complex, in which plant domestication might have taken place. 

Though I see a lot of value in this research, I do think the authors could improve and clarify a few things. First, they seem to imply they are theory building, which they are not. What they do is develop a model derived from general theories of gene-culture co-evolution and, perhaps to some degree, niche construction theory (NCT). To this end, I think the connections of their model to general theory could be made more explicit, particularly in the background/literature review. Research by Pete Richerson, Robert Boyd, Bruce Winterhalder, Melinda Zeder, and the like is relevant in this regard. In this context, it’s worth looking at the sometimes vehement debate between NCT theorists like Zeder and more behavioral-ecology (HBE) minded theorists like Piperno (both of whom are cited in the MS) and whether or not they are really all that alike, or if they are different (see lines 28-33). In short: though I see opportunity to marry these two perspectives, most researchers see them as opposed, in part due to the way they see agency playing out (see line 34). In fact, I see Angourakis and colleagues’ model articulating quite well with both HBE and niche construction advocates (worth looking at chapters in Kennett and Winterhalder’s “Behavioral Ecology and the Transition to Agriculture” here).

Authors’ reply: 

We disagree on not considering our work as theory building. The model is indeed a formalisation of a mechanism proposed as a general explanation of the phenomenon that is original in its formulation. It is not a direct implementation of a verbally explicit or formal model, yet clearly, our intention was to ground it in many well-established concepts in general biological theory (see Table 4 and 5).

We do appreciate and sympathise with most of the work done under other frameworks, including that of the authors mentioned by the reviewer, as well as both “sides” of the NCT vs HBE split. As the reviewer points out, we do present this paper with the hope that it can help reconcile the positions taken by NCT and HBE, but also attract the interest of those archaeologists and anthropologists that remain sceptical about the value of biological theory in this subject.

While our approach is aligned with some elements of gene-culture coevolution and niche construction theories, we did not derive the model from any of their contributions. As we mention in the first paragraph presenting the model (lines 64-67), the model is conceptually inspired by other contributions, such as the work of David Rindos, and takes the perspective of population ecology rather than population genetics. As the confusion is understandable, we add a clarification about this to the paragraph in line 57 (old manuscript):

> “The current work explores hypotheses on plant domestication and the origin of agriculture by using a coevolutionary framework capable of accounting for both plant and human factors. Our model combines readily-available formal models for mutualism and evolution used in population ecology, sociology and economics. Despite sharing the term "coevolution", our approach is neither based on nor necessarily aligned with the gene-culture coevolution or dual inheritance theory. The latter concerns a coupled process of genetic and cultural change in the same population and species, typically humans and other primates, in which other populations and species, and their changes, are considered as factors rather than the subjects of coevolution~\\cite{Feldman1996}. Likewise, the model we propose can be distinguished from human behaviour ecology models in this field since these have been defined in terms of human behaviour only (e.g., focusing on decision-making criteria) while factoring other species primarily as resources~\\cite{kennett_behavioral_2006,winterhalder_population_1988}.”

We have integrated more references from the HBE perspective, including Kennet & Winterhalder (2006), and further contextulise our position in relation to previous contributions:

> Replacing lines 29-30: “Approaches developed within human behavioural ecology ~\\cite{Piperno2017a,Smith2007,smith_general_2011,Rowley-Conwy2011,Laland2017,stiner_are_2016,kennett_behavioral_2006}, such as niche construction or cultural niche construction theories, have gained momentum in this effort.”

> Replacing lines 50-53: “Exceptionally, there have been key contributions from niche construction and optimal foraging theory as well as complex adaptative systems, but such contributions have been mostly centred on the human side of the process~\\cite{kennett_behavioral_2006,Freeman2012a,Freeman2015,Ullah2015,Brock2016}. Few simulation models have considered coevolution as the core mechanism producing changes in both plants and humans~\\cite{ullah_agmodel_2015,zhang_overview_2020}, while the first proposals in this line date back to almost fourty years ago~\\cite{Rindos1984}.”

However, given the already large size of the manuscript, we have actively avoided including a bibliographic review of the field, which, although relevant, is not the main subject of attention of the work summarised in the paper. Indeed, reviews on the subject can be found in other published works already cited in the manuscript where these different theoretical approaches have been extensively discussed. 

Second, particularly in the discussion section, but also in the paper more generally, the authors treat “Neolithisization” as a discrete, stage-like event, a result, at least in part, of a cursory presentation of non-agricultural (i.e., hunter-gatherer) behavioral diversity. A lot of current thinking on hunter-gatherers and their relationship to agriculturalists is that most or all of the behaviors necessary for agriculture—generation of surplus, storage, landscape management, concepts of private property, etc. are found in the diversity of hunter-gatherer behavior. I’d encourage the authors to look more closely at this literature and reframe their general approach and discussion to recognize that most of the behaviors we associate with agriculturalists, with the exception of domesticated plants, were likely present in many nonagricultural societies well before domestication. This could change their interpretation of their model’s results as well

Authors’ reply:

We agree with the reviewer that many behaviours associated with agriculture are also present in other types of subsistence strategies, including in many societies considered hunter-gatherers. However, we consider that the quantitative difference between the frequencies of these behaviours under the label of “agriculture” carries a qualitative separation to be made analytically. Thus, the use of the term agriculture, rather than intensive and managed plant gathering.

This said, our model does not, in fact, assume a categoric difference between human populations with or without human-plant coevolution, at least not beyond the variables defined (population size, frequency of types). Even more, our model assumes that all human types exist prior to the changes caused by coevolution, i.e., even the behaviours most favourable to plants are present in some measure in most simulation end-states.

We do not agree that we treat neolithisation as a stage-like process. Both the conceptual model and our implementation in R state continuous and non-linear dynamics. The classification of trajectories end-states in Results is made only on the basis of the coevolution coefficients, which express the distribution among types in each population. Furthermore, we highlight the diversity of scenarios within each group, depending on the conditions expressed by parameters, and mention that coevolved and not coevolved human populations under different conditions can potentially be confounded, if observed only in terms of population size (lines 274-278, 416-423). Last, we also discuss the potential of understanding multiple trajectories under a view of human-plant mutualism as a spectrum (lines 485-499), reinforcing the idea that there is a great diversity of scenarios prone to the development of agriculture, and that there is no single explanation that fits for all contexts to explain the emergence of agricultural systems.

One note, the mention of bioarchaeology in the abstract seems odd, as the authors themselves attest to the wide range of different studies outside of bioarchaeology that attest to the chronology of domestication and the development of agriculture.

Authors’ reply:

We agree with the point made. In a response also to Reviewer #2, we replaced the mention of “bioarchaeology” with “different branches of archaeology”.

On a final note, it was hard to tell which figures were being referred to in the text due to an absence of figure numbers (and corresponding figure captions). I think I eventually figured it out, but some good figure captions would really help the reader connect the text to the images.

Authors’ reply:

We are sorry about this, and we have now correctly referred to the figures in the text. See also the general note introducing our rebuttal.

Reviewer #2

The model presented here is useful and makes a good contribution to pushing forward theory on the mutualistic pathway of plant domestication. I commend the efforts to make the code available and the creation of a relatively user-friendly shiny app! 

However, a major issue with the code and how it is decribed in the text is the use of non human-readable variable names throughout. This greatly complicates the ability to read, understand, and reuse the code, and makes it very difficult to follow the narrative of the model logic in the paper. I strongly, strongly recommend that these variable be renamed using the accepted readable variable naming conventions that are common in open/reproducible code (e.g. CamelCase or underscore_case with short but descriptive variable names such as "InitialHumanPopulation" or "plant_type_n_coevolution_coefficient"). Yes, the variable names are longer, but they are actually readable and make it much easier to understand (and thus critique) the code and the narrative of the model function.

Authors’ reply:

We appreciate the reviewer’s suggestion and also value the importance of code readability. We have revised the manuscript, associated code, and all other related materials, changing all names accordingly. All variables and parameters (names mentioned directly in the manuscript) are now written in underscore_case (e.g., utility_per_capita_type_n_plants_to_humans) while extra names in code were revised as follows:

#### CODE STYLE NOTES: ####

# - All functions names with verbs

# - Main (public) functions inside the main pseudo class "hpcModel" using dot (.) as separator

# - higher-level composite objects in uppercase

# - function arguments using underscore (_) as separator

# - local variables and functions using camel case

So that equations can still be displayed as one-liners, we kept the mathematical notation as an alternative reference to variables and parameters. To improve the readability of equations, we now express all as they are, instead of using the generalised forms referring to populations A and B.

We released a new version (v1.3) of the repository with the source code and other materials and updated the reference used in the manuscript. We now mention both the Zenodo publication and the direct link to the GitHub source repository.

> “The source files associated with the HPC model are maintained in a dedicated online repository~\\cite{angourakis_andros-spicahpcmodel_2022}: \\url{https://github.com/Andros-Spica/hpcModel}. This repository contains several additional materials, including a web application to run simulations and the full report on the sensitivity analysis.”

angourakis_andros-spicahpcmodel_2022: Angourakis A, Alcaina-Mateos J. Andros-Spica/hpcModel: Human-Plant Coevolution model: source files, simulation interface, sensitivity analysis report and documentation; 2022. Available from: https://zenodo.org/record/6759456.

There also seemed to be an issue with the in-text figure numbering, which made it difficult to follow the results section. 

Authors’ reply:

We are sorry about this, and we have now correctly referred to the figures in the text. See also the general note introducing our rebuttal.

I have a few citations I'd like to see added as well, so I am attaching a marked up version of the manuscript with specific recommendations. 

Authors’ reply:

We have now added the recommended references and are grateful to Reviewer 2 for the suggestions.

I think the substantial conclusions and discussion are very good, with a few minor suggestions for how to extend these (again, in the attached marked up PDF). 

Authors’ reply: See comments and replies below.

In general, I think this is a highly effective and important work, and deserves to be published with these minor revisions. Also, I am happy to chat about any of this and don't care to remain anonymous -- feel free to send me an email! iullah@sdsu.edu

Comments added to the manuscript PDF

(line 14): Also, the record is fragmentary and we have only recovered and studied a very small portion of the small portion that has actually been preserved.

Authors’ reply:

We agree, the sentence is now: 

> “Domestication and agriculture emerged from diverse historical contexts and the empirical record available is manifold, inherently biased and fragmentary due to preservation issues, and it can often also be contradictory in evidencing causality~\\cite{Asouti2013}.”

(line 17): True, BUT, ethnoarchaeology is still an interesting and valid way to approach modeling. I would hesitate to throw the baby out with the bathwater here.

Authors’ reply:

Indeed, it is not our intention to discard those potential sources. We rephrased the part:

> “Furthermore, several models rely on ethnographic observations of contemporary traditional practices among indigenous peoples around the world~\\cite{Denham2004,Erickson2006,Gage2009,McCorriston1994,Roscoe2009}. While these practices make a useful basis for creating models of the past, they may greatly differ in context from those of the first communities engaging in agriculture within any given region, and therefore such "parallelisms" need to be used with care~\\cite{cunningham_perils_2018}.”

(line 52-53): Not to toot my own horn here, but check out: 1. Ullah IIT, Kuijt I, Freeman J. Toward a theory of punctuated subsistence change. PNAS. 2015;112: 9579–9584. doi:10/f7mv47

Authors’ reply:

We agree that the reference is relevant here, although we believe it even better fits with the previous sentence because it is where we discuss the human side of the process. We suggest the following change: 

> “Exceptionally, there have been key contributions from niche construction and optimal foraging theory as well as complex adaptative systems, but such contributions have been mostly centred on the human side of the process~\\cite{kennett_behavioral_2006,Freeman2012a,Freeman2015,Ullah2015,Brock2016}.”

(see also other changes in this paragraph, in reply to other suggestions given as comments in the manuscript PDF)

Moreover, we added the reference to line 27, as it also makes the call back to theory to explain the diversity of trajectories:

> “The analysis of this massive and relatively recent volume of data makes clear that it is now necessary to return to theory by revisiting the mechanisms allegedly involved in domestication, disentangling their connection to a diversity of trajectories~\\cite{Ullah2015,ahedo_lets_2021}, being those protracted or sudden, and identifying the weight of the social and ecological parameters.”

(line 61): I wonder if there is a more straightforward way of indicating that this is a purely theoretical model with simplified internal components?

Authors’ reply:

We believe that Lines 59-62 clearly state that this is a theoretical model, in opposition to data-driven approaches. However, we have complemented it slightly:

(line 59): “Our contribution is theoretical and explorative, thus it is not driven by the use of any specific dataset or case study. Furthermore, it does not carry the pretence —at least in its current form— of direct applicability to the many formats of empirical data.”

(line 74): I think it is worth somewhere mentioning here that this modeling method, although relatively novel, has been attempted before in the context of plant domestication. Specifically, this one: https://github.com/isaacullah/AgModel and this one: http://computationalsocialscience.org/wp-content/uploads/2016/11/CSSSA_2016_paper_39-1.pdf. A very brief comparison of this new model to the two existing models would be interesting somewhere in here.

Authors’ reply:

We add the references above to the manuscript, yet placing them still in the Introduction:

> Replacing lines 50-53: “Exceptionally, there have been key contributions from niche construction and optimal foraging theory as well as complex adaptative systems, but such contributions have been mostly centred on the human side of the process~\\cite{kennett_behavioral_2006,Freeman2012a,Freeman2015,Ullah2015,Brock2016}. Few simulation models have considered coevolution as the core mechanism producing changes in both plants and humans~\\cite{ullah_agmodel_2015,zhang_overview_2020}, while the first proposals in this line date back to almost fourty years ago~\\cite{Rindos1984}.”

As these references are good examples of bottom-up modelling on this subject, we also mention them in the Discussion (modifying lines 602-609):

> “As other examples in the past~\\cite{Rindos1984,winterhalder_population_1988}, the HPC model demonstrates that population-level (top-down) theory can still produce useful insights. Strong explanatory frameworks can be achieved without the fine insights of case-wise detail; an approach often resisted by archaeologists, but which is at the same time accepted whenever data is interpreted. In this sense, we consider that formal models are fundamental tools to present, demonstrate and explore any theoretical proposal. The HPC model also offers a solid basis for the design and further development of generative (bottom-up) models~\\cite{Epstein2006,cotto_nemo-age_2020,zhang_overview_2020,zhou_origin_2016,ullah_agmodel_2015}, and is complementary to approaches focusing on plant domestication syndrome through phenotypic and genetic characterisation~\\cite{Milla2015,Denham2020}.”

We think that a comparison of methodologies would yield a very interesting discussion, but believe that it would make the manuscript even more complex, moving the readers’ attention away from the intended scope. However, we do look forward to exploring model comparisons in the future, as we also believe it to be a necessary step towards model-based science in this field.

(table 1): I understand that there are a lot of variables here, but I want to point out that the variable names you have chosen are not "human readable," which makes your code much less accessible and/or reuseable. Here's a good recent reference: https://arxiv.org/abs/2109.10387 Variable names should be human readable, concise, and descriptive, using e.g CamelCase or underscore_case to concatenate words. Here's a good guide with explanation of why this is essential for open science and reproducible research: https://www.earthdatascience.org/courses/intro-to-earth-data-science/write-efficient-python-code/intro-to-clean-code/expressive-variable-names-make-code-easier-to-read/

(line 123): Your schematic figure of the model structure should help to follow this section, but with so many similarly named variables (and with variable names not in a particularly "human readable" format), it is exceedingly difficult to follow through the model logic using the schematic and the equations as written in this section. As such, I am still uncertain if I am correctly following the modeling logic, since I am mainly relying on the narrative description to assess if it is logically consistent.

Authors’ reply:

We reply to this in the first paragraph of our reply to Reviewer #2.

(line 300): I have noticed in some of my own modeling that the angle of the population line is sometimes variable when flipping between these states. In other words, the time between initialization of the coevolutionary pathway and completion of the coevolutionary process is different under different scenarios. It's hard to tell if this is the case in your model output because the x axis is very compressed in your Figure 3, but I wonder if you can check on this? Under which scenarios does coevolution proceed quickly, and when does it go slowly? This is particularly interesting in the context of the mounting evidence for a very long period of "pre-domestication" cultivation in many places around the world.

Authors’ reply:

We comment on the timing of change later in the text (see Parameter explorations). The conditions for delaying successful coevolution are shown in comparative terms in the more complex plots (e.g. Fig 8 and also in the HTML report on the sensitivity analysis experiments). We added a note about the timing of changes to Fig 3 caption:

> “Examples of trajectories and end-states produced by the Human-Plant Coevolution model. A: no coevolution; B: only plant population changes (domestication without cultivation); C: only human population changes (cultivation without domestication); D: some change happens in both populations (diverse populations); E: strong change in both populations (domestication and cultivation). More details on the timing of changes are given in the following sections.”

Reviewer #3

First, it's not clear to me what this research was intended to achieve. The closest we get to this is in lines 545-7: "...we aim to show how the succession of mixed economies are intrinsic parts of coevolutionary dynamics between humans and plants, and illuminate why these culminated, in many cases, in the origins of agriculture." If this was indeed the aim then, despite claims to the contrary (see below), the paper failed to achieve this.

Authors’ reply:

Our research explores the mechanism through which domestication of plant species and the development of agricultural economies could have taken place. As mentioned at the beginning of the manuscript, archaeological research needs to delve into the how/why (line 9) to understand the socio-ecological process leading to the emergence and consolidation of agricultural practices. The main objective of our research is to explore the selective pressures that arise in the interaction between two populations: one of humans and one of a non-determined plant species. We do this by using a modeling approach, and more specifically using the HPC model (see also lines 80-81).

Rather than addressing the 'how' and 'why' questions promised in the abstract and introduction, the paper is more descriptive than analytical, nor was it clear how the HPC model could be used in the future to address how and why questions concerning individual cases studies or in search of some common or overarching similarities in the trajectories to agriculture in different contexts.

Authors’ reply:

The paper describes the behaviour of the HPC model as a preliminary to exploring the dynamics of interaction between a hypothetical plant population and a human population. Such a detailed description is needed to clearly unfold the basis onto which the model was built and in which way it works (the model is sufficiently complex to need such a detailed description). Once we have done that, we explore the outputs of the model in relation to the human/plant dynamics under different settings, and we group and discuss the outputs into three different groups. Such discussion tries to clarify the “how and why” there has been the emergence of agriculture in some parts of the world (see a synthesis starting from line 246, where we discuss these different end-states). 

I therefore found the conclusions drawn towards the end of the paper unsubstantiated by the research as presented. For example: the conclusions that the HPC model “can greatly help understand…the origins of agriculture” or “can produce useful insights” or “offers a solid basis for the development of generative models” – where is the evidence for this in the paper? How would the model achieve such objectives? Greater clarity of purpose and a more realistic account of achievement is needed.

Authors’ reply:

As clearly argued in the manuscript, the model defines a general mechanism able to frame the interactions between human societies and plant species, yielding three groups of possible scenarios. These scenarios include the absence of coevolution and the consolidation of a weak mutualistic ecological relationship (and therefore no steps towards domestication), while many other scenarios show partial and complete coevolutionary dynamics, some of which would be characterised as the emergence of an agricultural system. Such different scenarios (no coevolution, partial or complete coevolution) are the expression of possible realities arising from the two species' interactions, and they can be used to explain the change from plant cultivation to domestic plant agriculture. The formalisation of a double positive feedback loop mechanism (coevolution of populations engaging in mutualism) is fundamentally the key to understanding such processes and, we argue, offers a framework for other modelling and theorising approaches.

We have discussed these points in the manuscript, and we believe that we have been sufficiently unambiguous in arguing how the proposed approach can “greatly help understand…the origins of agriculture” and how it “can produce useful insights”. As many other contributions in the past have shown, by making experiments in silica, we can explore the dynamics of a process that otherwise would be inaccessible through empirical experimentation. For example, we can discern and relatively quantify the significance of different behavioural/phenotypic traits in humans and plants for the emergence of agriculture as a system. The explicitation of the premises onto which the HPC model was built, “offers a solid basis for the development of generative models”, as it suggests a more systematic, yet general, schematic for the representation of human and plant agents.

---

## [Editor Report · Decision Letter 1]

22 Jul 2022

Human-Plant Coevolution: A modelling framework for theory-building on the origins of agriculture

PONE-D-21-36515R1

Dear Dr. Angourakis,

We’re pleased to inform you that your manuscript has been judged scientifically suitable for publication and will be formally accepted for publication once it meets all outstanding technical requirements.

Kind regards,

Raven Garvey, Ph.D.

Academic Editor

PLOS ONE
---

## [Editor Report · Acceptance letter]

16 Aug 2022

PONE-D-21-36515R1 

Human-Plant Coevolution: A modelling framework for theory-building on the origins of agriculture 

Dear Dr. Angourakis:

I'm pleased to inform you that your manuscript has been deemed suitable for publication in PLOS ONE. Congratulations! Your manuscript is now with our production department. 

Kind regards, 

on behalf of

Dr Raven Garvey 

Academic Editor

PLOS ONE